# ANCHOR-MoE: A MEAN ANCHORED MIXTURE OF EXPERTS FOR PROBABILISTIC REGRESSION

## ABSTRACT

We present **Anchor–MoE**, an anchored mixture-of-experts for probabilistic and point regression. A base anchor prediction is concatenated with the inputs and mapped to a compact latent space. A learnable metric window with a soft top-$k$ router induces sparse weights over lightweight MDN experts, which output residual corrections and heteroscedastic scales. Training uses negative log-likelihood with an optional held-out linear calibration to refine point accuracy. Theoretically, under Hölder-smooth targets and fixed partition-of-unity weights with bounded overlap, Anchor–MoE attains the minimax-optimal $L^2$ rate $N^{-2\alpha/(2\alpha+d)}$. The CRPS generalization gap is $\tilde{\mathcal{O}}\big(\sqrt{(\log(Mh) + P + k)/N}\big)$ under bounded-overlap routing, and an analogous scaling holds for test NLL under bounded moments. Empirically, on standard UCI benchmarks, Anchor–MoE matches or surpasses strong baselines in RMSE and NLL, achieving state-of-the-art probabilistic results on several datasets. Anonymized code and scripts will be provided in the supplementary material.

## 1 INTRODUCTION

Regression is a cornerstone of machine learning: given covariates $\mathbf{X}$ and a real-valued response $Y$, the goal under mean squared error(MSE) loss is to estimate the conditional expectation $f^\star(x) = \mathbb{E}[Y \mid \mathbf{X} = x]$, which is the population risk minimizer. Regression methods are ubiquitous in modern research, powering applications from climate forecasting Chau et al. (2021) and protein engineering Michael et al. (2023) to chronic disease prognosis Zhang et al. (2023).

Most machine learning approaches cast regression as learning a deterministic mapping and optimize mean-squared error, effectively estimating $\mathbb{E}[\mathbf{Y} \mid \mathbf{X}]$. However, Kendall and Gal Kendall & Gal (2017) show that explicitly modeling the full predictive distribution, especially heteroscedastic noise—can improve point accuracy by weighting residuals with learned uncertainty. In this probabilistic regression view we learn $p(\mathbf{Y} \mid \mathbf{X})$ rather than only its mean, enabling calibrated uncertainty quantification and better downstream decisions (e.g., financial risk management), with strong empirical

Building on these practical benefits, a range of probabilistic regression families has been proposed. Kendall & Gal (2017) develop uncertainty–aware neural approaches for probabilistic regression; Seiller et al. (2024) propose tree–based probabilistic ensembles; Rigby & Stasinopoulos (2005) formalize distributional generalized additive models (GAMLSS) that model location, scale, and shape. While all return full predictive distributions, they involve different trade–offs: deep and ensemble methods can be computationally intensive and often reduce interpretability; GAMLSS requires specifying the response distribution and link functions and can be challenging to scale in very high–dimensional settings; and, in practice, some probabilistic models may favor calibration over point accuracy on certain datasets.

Several recent works have sought to address these limitations. Hu et al. (2019) propose a neural architecture that outputs a full predictive density in a single forward pass, substantially reducing computation for deep probabilistic models. Zhang et al. (2020) develop an *Improved Deep Mixture Density Network* (IDMDN) for regional wind-power probabilistic forecasting across multiple wind farms, demonstrating robust accuracy in high-dimensional settings. Rügamer et al. (2023) blend classical structured statistical effects with deep neural networks via semi-structured distributional regression, enabling flexible modeling that accommodates both tabular and image data. Finally,

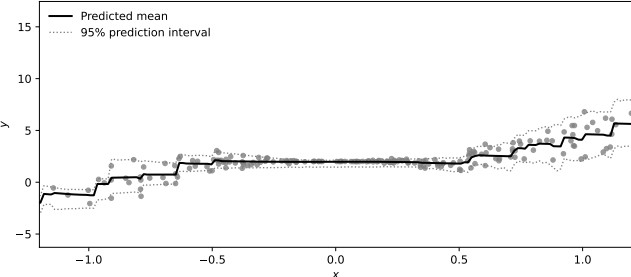

Figure 1: **Anchor–MoE overview.** (*i*) A base regressor produces the anchor from the same inputs, we concatenate inputs and the anchor, then project to a latent $z$, score locality via a learnable metric–window, and apply soft top-$k$ routing to a few MDN experts. (*ii*) Experts output mixture parameters; the weighted mixture yields the predictive density $p(y \mid x)$ used for probabilistic metrics.(*iii*) For point accuracy, we use least square method to calibrate the mean.

Martin Vicario et al. (2024) present an uncertainty-aware deep-learning pipeline that assigns reliability scores to predictions based on quantified uncertainty, enhancing interpretability in safety-critical applications. Collectively, these advances have helped push forward probabilistic regression and uncertainty estimation.

Recently, Duan et al. (2020) introduced Natural Gradient Boosting (NGBoost), which fits the parameters of a chosen predictive distribution by boosting decision–tree base learners with natural–gradient updates. NGBoost is competitive on many tabular benchmarks with relatively little tuning, making it simple to deploy. However, several limitations arise in regression settings. First, NGBoost requires the user to pre–specify a parametric base distribution, and accuracy can degrade under misspecification. Second, the original formulation is univariate; for multivariate targets one must either train separate models or adopt an extension that models joint uncertainty. While O'Malley et al. (2021) extend NGBoost to multivariate outputs by learning a joint distribution, this increases computational cost and implementation complexity. Finally, beyond general boosting theory, the original work offers limited task–specific statistical guarantees.

Mu & Lin (2025) demonstrate that the mixture-of-experts(MoE) model can better fitting the heterogeneous and complex data with less computational resources. Based on that, We propose Anchor–MoE, a simple two-stage, modular architecture for probabilistic and point regression to overcome above challenges. For Anchor-MoE, Stage 1 uses a small tuned gradient-boosted trees (GBDT) model to produce an anchor mean $\hat{\mu}_a(x)$. Stage 2 concatenates the standardized anchor to the inputs and projects to a compact latent space; a learnable metric–window kernel together with a soft top-$k$ router yields sparse weights over $K$ lightweight mixture-of-density networks(MDN) experts. Experts output a small Gaussian mixture. In the default the anchor predicts a residual on top of the anchor and a variance. Training minimizes NLL with mild entropy regularization, that is, we augment the NLL with a tiny entropy, see details in appendix. A disjoint calibration split fits a linear map on predicted means to improve RMSE we report RMSE on calibrated means and NLL on the uncalibrated $z$-space density. The design is plug-and-play, see Figure 1 for an overview.

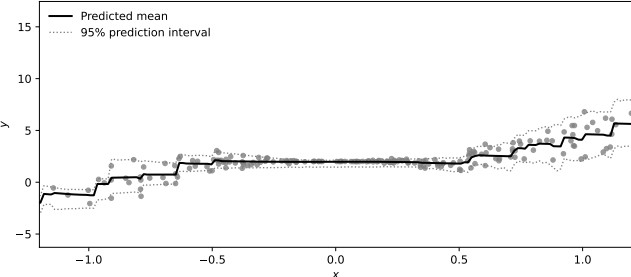

Figure 2: Interval predicted by Anchor-MoE on 1-dimensional toy probabilistic regression problem. Dots represents the data points. Black line is predicted mean and gray lines are upper and lower 95% covered distribution predicted.

## 2 METHOD

In this section, we will introduce and analyze each component of our model and give a default configuration which is used for experiment part at the end of this section.

In standard prediction settings the object of interest is a scalar function such as $\mathbb{E}[Y \mid X = x]$. In probabilistic regression we instead aim to learn a full predictive law $P_{\Theta(x)}(y \mid x)$. Our approach is to parameterize $P_{\Theta(x)}$ by a mixture family whose parameters $\Theta(x)$ are smooth functions of the input.

Concretely, Anchor–MoE first forms a strong anchor mean $\mu_{\mathrm{a}}(x)$ using a small gradient–boosted tree. The anchor is concatenated to the features and mapped to a compact latent space, from which a metric–window router produces sparse (soft top-$k$) mixture weights. Each activated expert is a lightweight MDN that predicts a local residual $\delta$ to the anchor and a scale, so that the resulting predictive distribution is a mixture with means $\mu_{\mathrm{a}}(x) + \delta$ and heteroscedastic variances.

The next subsections detail the components: The latent projection and metric window (Section 2.2), the latent metric–window and router (Section 2.3), the expert MDN heads and training objective (Section 2.4), and the post-hoc mean calibration (Section 2.5).

## 2.1 BACKGROUND AND NOTATION

We consider i.i.d. samples $(x, y)$ with $x \in \mathbb{R}^d$ and $y \in \mathbb{R}$. A probabilistic regressor specifies a conditional law $p_\theta(y \mid x)$ with predictive mean $\mu(x)$ and variance $\sigma^2(x)$. We evaluate with the average negative log-likelihood (NLL) on a test set $\{(x_i, y_i)\}_{i=1}^n$,

$$\mathrm{NLL} \;=\; \frac{1}{n} \sum_{i=1}^n \big[ -\log p_\theta(y_i \mid x_i) \big].$$

We also report the continuous ranked probability score (CRPS) Gebetsberger et al. (2018), defined for a predictive CDF $F(\cdot \mid x)$ as

$$\mathrm{CRPS}\big(F(\cdot \mid x), y\big) = \int_{-\infty}^{\infty} \big( F(t \mid x) - \mathbf{1}\{y \le t\} \big)^2 \, dt.$$

In practice we use the standard closed-form for Gaussian mixtures.

An external anchor $a(x)$ is a strong point predictor trained on the train/validation split. We use it in two roles: (i) as an additional feature by concatenation of inputs and the anchor mean, and (ii) as a residual reference so that expert means correct $a(x)$ by a learned $\Delta(x)$.

We map the concatenated data to a $D$-dimensional latent code $z$ via a linear projection and normalization. A learnable metric window together with a soft top-$k$ router produces weights $\alpha(z)$ over $K$ experts. Each expert outputs a small $C$-component Gaussian mixture with weights $\pi_{j,c}(x)$, means $\mu_{j,c}(x)$, and scales $\sigma_{j,c}(x) > 0$. The predictive density is a mixture

$$p_\theta(y \mid x) \;=\; \sum_{j=1}^K \alpha_j(z) \sum_{c=1}^C \pi_{j,c}(x)\, \mathcal{N}\big(y; \mu_{j,c}^{\mathrm{eff}}(x), \sigma_{j,c}^2(x)\big),$$

where in residual mode $\mu_{j,c}^{\mathrm{eff}}(x) = a(x) + \Delta_{j,c}(x)$, and in free-mean mode $\mu_{j,c}^{\mathrm{eff}}(x) = \mu_{j,c}(x)$. Since the model consists of many parts and each part has numerous hyper-parameters, we report a summary table 1 of key hyper-parameters to the structure more clearer.

Table 1: Key hyper-parameters for each part

| Module | Key hyper-parameters |
| --- | --- |
| Anchor | n_estimators, learning_rate, max_depth, subsample; select best_iter on validation |
| Projection / Latent | latent dimension $D$; normalization on/off; weight decay $\lambda$ |
| Metric window | number of experts $K$; scale clamp $[\tau_{\min}, \tau_{\max}]$; window L2 $\lambda_{\mathrm{win}}$ |
| Router | top-$k$ ($k$); temperature $\tau$; smoothing $\varepsilon$; load-balance coefficient $\lambda_{\mathrm{lb}}$ |
| Mixture of Experts (MDN) | width $h$ and depth $L$; components $C$; $\sigma$ clamp $[\sigma_{\min}, \sigma_{\max}]$ |
| Calibration | calibration split size; linear map parameters $(a, b)$ |

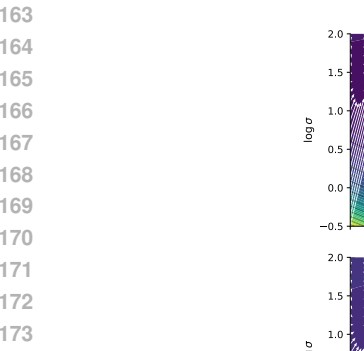

Figure 3: Loss landscapes and gradient fields for learning a normal distribution. Each panel shows the score surface in the $(\mu, \log \sigma)$ plane with its gradient vectors. The landscapes of NLL and CRPS are identical up to a monotone transform, so both are proper and target the same optimum. The difference lies in the gradients: without an anchor (top), the NLL field shows strong coupling between $\mu$ and $\log \sigma$, yielding slanted directions that can cause zig-zagging and early shrinkage of $\sigma$. With Anchor+$\Delta$ (bottom), the parameterization recenters the mean around the anchor and reduces mean–scale coupling; gradients become closer to axis-aligned and the path to the optimum is more stable. The CRPS field is also smoother in the tails than NLL, leading to milder updates of $\sigma$ when $\mu$ is off target.

## 2.2 LATENT PROJECTION AND METRIC WINDOW

We map the input $x$ to a $D$-dimensional latent code $z$ using a linear projection followed by LayerNorm.

Locality is scored by a learnable metric window. Each expert $j$ has a center $c_j$ and a positive scale vector $s_j$. The unnormalized score is

$$\tilde{w}_j(z) \;=\; \exp\!\Big(-\tfrac{1}{2} \left\| (z - c_j) \odot s_j^{-1} \right\|_2^2\Big),$$

For stability, we clamp all log-scales to a fixed range and add a small $\ell_2$ penalty on the log-scales.

To obtain sparse and robust routing, we keep the $k$ largest entries of $w(z)$ and renormalize within this active set. During training, we apply a tiny smoothing $\varepsilon$ within the active set to avoid zero gradients; the same top-$k$ rule is used at inference.

## 2.3 ROUTER

We follow the classic gating view of MoE Jacobs et al. (1991); Jordan & Jacobs (1994) and combine a lightweight content router with the metric window.Given the latent code $z$, we form a query $q = W_q z$ and maintain keys $\{k_j\}_{j=1}^K$ in $\mathbb{R}^{d_r}$. We use scaled dot–product logits with temperature $\tau$ (cosine normalization is optional):

$$\ell_j(z) \;=\; \frac{\langle q, \, k_j \rangle}{\sqrt{d_r} \, \tau}.$$

We fuse the router with the locality weights $w(z)$ by simple multiplication, then renormalize:

$$\alpha_j(z) \;\propto\; w_j(z) \, \mathrm{softmax}(\ell(z))_j, \qquad \sum_j \alpha_j(z) = 1.$$

For specialization, we keep the $k$ largest entries of $\alpha(z)$ and renormalize within this active set. During training, a tiny smoothing $\varepsilon$ is applied within the active set to avoid zero gradients; at inference we use

the same top-$k$ rule without smoothing. This router adds $\mathcal{O}(Kd_r)$ work per example and suppresses far-away experts while enabling content-dependent gating.

## 2.4 Mixture of Experts

MDN model the full conditional distribution and suit heteroscedastic or multi-modal targets Bishop (1994). The window and the router produce nonnegative weights $\alpha_j$. Each expert is a small MDN with $C$ Gaussian components. Each expert outputs mixture weights via softmax, component means, and positive scales. Scales are clamped to a fixed range for numerical stability.

Anchor coupling. Three modes are supported: anchor+delta as the default, anchor only, and free. A small $\ell_2$ penalty on the residual discourages unnecessary drift. The anchor value is also concatenated to the inputs of the expert and the router.

Predictive density. For a univariate target

$$
p(y \mid x) = \sum_{j=1}^{K} \sum_{c=1}^{C} \alpha_j(x)\, \pi_{j,c}(x)\, \phi_{j,c}(y \mid x),
$$

where $\phi_{j,c}$ is a Gaussian density with mean $\mu_{j,c}^{\mathrm{eff}}(x)$ and variance $\sigma_{j,c}^2(x)$. This design lets experts specialize locally while the gates provide smooth interpolation.

## 2.5 Calibration

We hold out a small calibration split and fit a single affine map by least squares in z space: $\mu_{\mathrm{cal}} = a\,\mu + b$. At test time we apply this map to the model mean and report RMSE in original units. The predictive variance is left unchanged and we report NLL on the original uncalibrated density.

## 3 Theoretical Analysis

The analysis explains what each design choice controls and when gains should appear. It turns the architecture into testable statements that can be checked on data. The assumptions are built into the model: a bounded latent projection with clamped window scales gives smooth and stable locality scores; top-k routing limits the number of active experts per input; variance clamping in the latent space avoids degenerate likelihoods. From these ingredients the theory yields the following predictions.

1. With the variance clamp in place, lowering NLL should be accompanied by lower RMSE on the predictive mean.

2. At fixed $k$, increasing $K$ improves risk up to a knee point, after which gains become marginal as estimation error dominates.

3. Moving from $k = 1$ to $k = 2$ stabilises gating and often improves CRPS, with diminishing returns for larger $k$.

4. Light entropy on the gates and small scale regularisation improve load balance, reduce routing variance, and make training more stable.

5. Exposing the anchor by concatenation or by residual shift reduces mean bias, with larger benefits on datasets that show stronger input dependent noise.

## 3.1 Approximation and minimax−optimal rates

We assume the target regression function is Hölder–$\alpha$ smooth on a $d$–dimensional cube. A partition of unity with $K$ local windows and bounded overlap $k$ gives an interpolation error that decays with $K$:

$$
\text{approximation error} \;\asymp\; K^{-2\alpha/d}.
$$

Fitting $K$ experts from $N$ samples under overlap $k$ and per–expert capacity comp contributes an estimation term

$$\text{estimation error} \asymp \frac{k \operatorname{comp} K}{N}.$$

Balancing the two terms yields the usual choice

$$K^{\star} \asymp N^{d/(2\alpha+d)},$$

and the corresponding risk achieves the minimax rate

$$\mathbb{E}\Big[\,\|\,\widehat{f} - f^{\star}\,\|_{L^2}^2\,\Big] \lesssim N^{-2\alpha/(2\alpha+d)}.$$

In our setting the latent projection is bounded, window scales are clamped, and routing activates only $k$ experts. These design choices enforce the bounded–overlap and smoothness conditions used above, so the rate prediction is meaningful for the proposed model. We train with Gaussian NLL; because predictive variances are clamped away from 0 and $\infty$, lowering NLL also lowers the mean–squared error of the predictive mean up to constant factors. This is why we report both NLL and RMSE in the experiments.

## 3.2 GENERALISATION UNDER CRPS

CRPS is Lipschitz in the predictive cdf under the $L^1$ metric, and the loss is bounded once expert means and variances are bounded and the response is bounded. Write the bound as $|\mathrm{CRPS}| \le B$ with $B = R_f + R_y + \sqrt{2/\pi}\,\overline{\sigma}$, where $R_f$ bounds the expert means, $\overline{\sigma}$ bounds the standard deviation from above, and $R_y$ bounds the response.

Let $\mathcal{R}$ be the population CRPS risk and $\widehat{\mathcal{R}}_N$ its empirical counterpart on $N$ samples. For any $\delta \in (0,1)$, with probability at least $1 - \delta$,

$$\mathcal{R} - \widehat{\mathcal{R}}_N \le 4\,\mathcal{R}_N(\mathcal{F}) + 3B\sqrt{\frac{\log(2/\delta)}{2N}},$$

where $\mathcal{R}_N(\mathcal{F})$ is the empirical Rademacher complexity of the CRPS–induced function class.

Under mild size controls on the model, this complexity satisfies

$$\mathcal{R}_N(\mathcal{F}) \le C\sqrt{\frac{\log(Mh) + P + K}{N}},$$

with $M$ mixture components per expert, expert width proxy $h$, router size $P$, and number of experts $K$; $C$ is a constant independent of $N$. With top-$k$ bounded–overlap gating, the dependence on $K$ can be replaced by the active overlap $k$.

## 4 EXPERIMENTS

### 4.1 EXPERIMENTAL SETUP

We first run a light heteroscedasticity screening with OLS residuals to confirm input-dependent noise, then keep a single protocol across datasets. Following Hernández-Lobato & Adams (2015), we evaluate on nine UCI datasets with a 90%/10% train/test split; inside the training fold, 20% is held out to choose the number of boosting stages for the anchor by validation NLL, after which the chosen stage is refit on the full training fold and the MoE is trained on top. Each experiment is repeated 20 times and we report the mean and standard error. The anchor mean is concatenated to the inputs; a small disjoint calibration split fits a least-squares linear map on predicted means while leaving variances unchanged. Unless stated otherwise, we fix the configuration summarized in Table 2 and report NLL on the uncalibrated predictive density in z-scored space and RMSE on calibrated means in the original scale. For PROTEIN, we subsample 10,000 examples per run and retrain NGBoost on the same subsamples for fairness; for the remaining datasets we use all samples and cite NGBoost from Duan et al. (2020).

Table 2: Fixed configuration for experiments

| Component | Setting |
|---|---|
| Latent projection | Dimension $D=2$ |
| Experts | $K=8$ experts; each expert is an MLP of width 128 with an MDN head of $C=3$ components |
| Router | Top-$k$ gating with $k=2$ and light smoothing |
| Variance clamp | Predicted standard deviation clamped to $[0.05, 1]$ |
| Anchor model | Gradient-boosted trees; best iteration chosen by validation NLL and then refit on the full training fold |

## 4.2 HETEROSCEDASTICITY DIAGNOSTICS

Since prior work shows that learning input-dependent variance can be beneficial Nix & Weigend (1994); Kersting et al. (2007), we first check whether residual variance depends on the inputs before comparing probabilistic models, it should help most when noise varies with the covariates. We therefore run a light screening for heteroscedasticity on each dataset to contextualize the results.

We fit an Ordinary Least Squares(OLS) model and test input–dependent noise using standard diagnostics: Breusch–Pagan for linear variance in regressors Breusch & Pagan (1979), White's general test for heteroskedasticity White (1980), Goldfeld–Quandt along the fitted-value ordering Goldfeld & Quandt (1965), Levene's robust test across fitted-value bins Levene (1960), and a Spearman rank correlation between absolute residuals and fitted values Spearman (1904).In our analysis we treat the White test as the primary decision signal White (1980), with Breusch–Pagan and Levene used as corroborating evidence Breusch & Pagan (1979); Levene (1960).

Table 3: Heteroscedasticity diagnostics on UCI datasets; extremely small $p$–values reject homoscedasticity.

| Dataset | $N$ | $p_{BP}$ | $p_{White}$ | $p_{GQ}$ | $p_{Spearman[\epsilon]}$ | $|\rho|_{Spearman}$ | $p_{Levene}$ | R2_log_resid2 |
|---|---|---|---|---|---|---|---|---|
| Yacht | 308 | $1.006\,588 \times 10^{-12}$ | $4.513\,610 \times 10^{-18}$ | $2.957\,812 \times 10^{-155}$ | $2.621\,576 \times 10^{-11}$ | 0.368 | $3.948\,916 \times 10^{-35}$ | 0.117 072 |
| Energy | 768 | $4.981\,111 \times 10^{-62}$ | $1.134\,108 \times 10^{-100}$ | $5.950\,234 \times 10^{-54}$ | $2.233\,622 \times 10^{-13}$ | 0.260 | $9.763\,824 \times 10^{-40}$ | 0.047 926 |
| Protein* | 10000 | 0.000 000 | 0.000 000 | $1.351\,570 \times 10^{-183}$ | 0.000 000 | 0.301 | $7.939\,445 \times 10^{-205}$ | 0.046 301 |
| Concrete | 1030 | $9.204\,946 \times 10^{-26}$ | $8.574\,022 \times 10^{-38}$ | $1.757\,200 \times 10^{-37}$ | $6.057\,033 \times 10^{-18}$ | 0.264 | $5.597\,492 \times 10^{-24}$ | 0.041 329 |
| Wine | 1599 | $1.587\,975 \times 10^{-13}$ | $1.896\,220 \times 10^{-26}$ | $3.288\,921 \times 10^{-9}$ | $3.579\,995 \times 10^{-16}$ | 0.202 | $8.619\,177 \times 10^{-21}$ | 0.034 424 |
| Housing | 506 | $6.265\,431 \times 10^{-9}$ | $2.266\,143 \times 10^{-25}$ | $5.388\,709 \times 10^{-15}$ | $3.079\,593 \times 10^{-4}$ | 0.160 | $6.969\,489 \times 10^{-4}$ | 0.015 865 |
| Kin8nm | 8192 | $4.508\,460 \times 10^{-50}$ | $2.869\,093 \times 10^{-301}$ | $7.956\,050 \times 10^{-39}$ | $3.315\,300 \times 10^{-41}$ | 0.148 | $3.761\,213 \times 10^{-29}$ | 0.015 179 |
| Naval | 11934 | 1.000 000 | 0.000 000 | $2.346\,512 \times 10^{-3}$ | $3.442\,543 \times 10^{-33}$ | 0.110 | $4.532\,954 \times 10^{-77}$ | 0.004 240 |

All datasets reject homoscedasticity by the White test at the one percent level. Effect sizes differ: Yacht is large (Spearman $\approx 0.37$, $R^2 \approx 0.12$); Energy, Protein, Concrete, and Wine are moderate ($R^2 \approx 0.03$–$0.05$; Spearman $\approx 0.20$–$0.30$); Housing, Kin8nm, and Naval are small ($R^2 \le 0.016$; Spearman $\le 0.16$). Thus input–dependent noise is ubiquitous but uneven, and the expected gain from learning variances should be strongest where these effect sizes are larger.

EMPIRICAL EXPERIMENTS

We run empirical experiments informed by the heteroscedasticity screening. Uncertainty quality is evaluated with the average test negative log-likelihood, where lower values are better. The primary baseline is NGBoost, and results for additional baselines are in the Appendix. Although Anchor–MoE targets uncertainty estimation, a point prediction is obtained as the predictive mean, and we assess it with test RMSE. For RMSE we apply a small least-squares mean calibration on a disjoint split as described in Section 3.4, while NLL is computed on the uncalibrated density in z-scored space. Unless stated otherwise the configuration matches the uncertainty experiments. We use gradient-boosted trees as the default anchor for reproducibility, and other anchors can be substituted without changing the pipeline. To quantify the contribution of each component we run ablations under the same setup. In the default anchor plus delta mode a small boosted model produces an anchor mean, expert heads learn residuals that correct this anchor and also output variances. In No-Anchor we remove the anchor feature and the residual coupling so experts predict free means. In No-Router we disable the dot-product router and rely only on the metric window with the same top-k mask and smoothing, then renormalize. In No-Cal we compute RMSE on uncalibrated means. Main comparisons to NGBoost and ablation outcomes are reported in Tables 5 and 4b.

Table 4: UCI benchmarks: test NLL (left) and RMSE (right). Best per row in **bold**.

(a) NLL. NGBoost numbers are from Duan et al. (2020); other baselines follow prior reports (see Table 8 in the appendix). Anchor–MoE is competitive on complex datasets.

| Dataset | N | Anchor–MoE | NGBoost |
|---------|------|-------------|----------|
| Boston | 506 | **0.60 ± 0.11** | 2.43 ± 0.15 |
| Concrete | 1030 | **0.25 ± 0.06** | 3.04 ± 0.17 |
| Energy | 768 | **-1.68 ± 0.20** | 0.46 ± 0.06 |
| Kin8nm | 8192 | 0.12 ± 0.01 | **-0.49 ± 0.02** |
| Naval | 11934 | -1.26 ± 0.02 | **-5.34 ± 0.04** |
| Power | 9568 | **-0.15 ± 0.02** | 2.79 ± 0.11 |
| Protein | 10000 | **1.06 ± 0.04** | 1.24 ± 0.04 |
| Wine | 1599 | **1.20 ± 0.02** | 4.96 ± 0.60 |
| Yacht | 308 | **-1.80 ± 0.04** | 0.20 ± 0.26 |

(b) RMSE. Anchor–MoE offers results comparable to NGBoost.

| Dataset | N | Anchor–MoE | NGBoost |
|---------|------|-------------|----------|
| Boston | 506 | 3.01 ± 0.14 | **2.94 ± 0.53** |
| Concrete | 1030 | **4.45 ± 0.16** | 5.06 ± 0.61 |
| Energy | 768 | 0.47 ± 0.02 | **0.46 ± 0.06** |
| Kin8nm | 8192 | **0.07 ± 0.00** | 0.16 ± 0.00 |
| Naval | 11934 | 0.00 ± 0.00 | 0.00 ± 0.00 |
| Power | 9568 | **3.21 ± 0.05** | 3.79 ± 0.18 |
| Protein | 10000 | **4.41 ± 0.02** | 4.44 ± 0.02 |
| Wine | 1599 | 0.62 ± 0.01 | **0.60 ± 0.01** |
| Yacht | 308 | 0.62 ± 0.06 | **0.50 ± 0.20** |

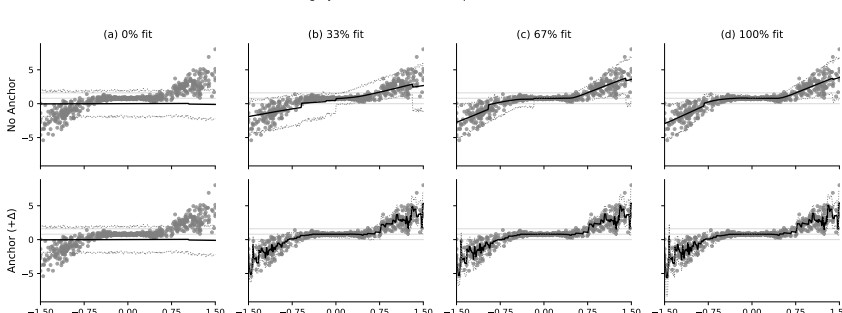

Figure 4: Learning dynamics on a toy 1D dataset: No-Anchor (top) vs Anchor ($+\Delta$, bottom) at 0%, 33%, 67%, and 100% fit. Line as in Figure 1. Without anchor, updates emphasize global trend and show larger oscillations with tail variance inflation; with anchor, updates are balanced, the central plateau is preserved earlier, and predictive intervals are better calibrated.

Table 5: Comparison on UCI Benchmark dataset as measured by NLL while ablating key components of Anchor-MoE. Bolding is as in Table 1.

| Dataset | N | Anchor-MoE | Anchor | Router | Calibration |
|---------|------|-------------|---------|---------|-------------|
| Boston | 506 | 0.60 ± 0.11 | 0.83 ± 0.24 | **0.51 ± 0.05** | 0.52 ± 0.05 |
| Concrete | 1030 | 0.25 ± 0.06 | 0.73 ± 0.04 | **0.20 ± 0.05** | **0.20 ± 0.06** |
| Energy | 768 | **-1.68 ± 0.2** | -1.30 ± 0.05 | -0.76 ± 0.05 | -0.96 ± 0.05 |
| Kin8nm | 8192 | **0.12 ± 0.01** | 0.68 ± 0.02 | 1.00 ± 0.01 | 0.97 ± 0.01 |
| Naval | 11934 | **-1.26 ± 0.02** | -1.09 ± 0.02 | -1.10 ± 0.02 | -1.12 ± 0.02 |
| Power | 9568 | -0.15 ± 0.02 | -0.05 ± 0.03 | -0.15 ± 0.02 | **-0.18 ± 0.02** |
| Protein | 10000 | 1.06 ± 0.04 | **0.63 ± 0.01** | 1.05 ± 0.02 | 0.90 ± 0.03 |
| Wine | 1599 | **1.20 ± 0.02** | 1.52 ± 0.43 | 1.16 ± 0.02 | 1.21 ± 0.03 |
| Yacht | 308 | -1.80 ± 0.04 | 0.24 ± 0.42 | -1.76 ± 0.03 | **-1.83 ± 0.03** |

## 5 CONCLUSION

We presented Anchor–MoE, a modular approach for point and probabilistic regression. A small tree model provides an anchor mean, a metric window with a soft top-k router dispatches inputs sparsely to mixture-density experts, scaling Anchor–MoE with expert sharding and switch-style routing, which is compatible with existing systems Lepikhin & et al. (2020); Fedus et al. (2021) and is a natural next step. And a one dimensional post hoc calibrator corrects mean bias. The parts are

loosely coupled, easy to ablate, and the same design can be adapted to classification or survival by changing the likelihood.

A central finding is the alignment between heteroscedasticity diagnostics and empirical gains. Datasets with strong input dependent noise such as Yacht, as indicated by very small test p values together with larger effect sizes in the simple $R^2$ on $\log(e^2)$ and in the absolute Spearman correlation, are exactly where Anchor–MoE delivers the clearest improvements in test negative log-likelihood and better interval behavior. On datasets with moderate signals such as Energy, Concrete, Wine, Power, and Protein, Anchor–MoE improves likelihood metrics consistently while keeping root mean squared error close to the best baseline; the anchor plus delta design lets experts spend capacity on local residuals and variance rather than relearning the global mean. When diagnostics point to weak heteroscedasticity as in Housing, Kin8nm, and Naval, the advantage in likelihood narrows or can reverse, and simple mean focused models can be sufficient for point accuracy. This pattern matches the intended role of the method: model uncertainty where noise truly varies with inputs, avoid unnecessary variance modeling when noise is nearly constant.

Ablations clarify mechanism. Removing the anchor pushes experts to absorb mean bias through variance inflation, which can reduce likelihood quality and harm coverage. Disabling the router removes content dependent specialization and leaves only the window kernel to gate, which consistently hurts likelihood and sometimes point accuracy on complex data. Removing mean calibration increases bias and worsens root mean squared error without a benefit to likelihood in the z scored space. Together, these results support the default of anchor concatenation and residual correction, soft top k routing with bounded overlap, and a light least squares mean calibration.

Theoretical guidance also matches practice. Bounded overlap routing and fixed expert capacity control estimation error, while the window partition controls approximation. Keeping a small number of experts and a small top k across datasets respects these capacity assumptions, and the observed stability across random splits is consistent with generalization bounds stated for continuous ranked probability score and with the link between negative log-likelihood and mean squared error under bounded variances. In short, the design choices used in the main tables are the ones that make the theory applicable.

For practitioners, a simple rule emerges from the diagnostics. If a quick screening rejects homoscedasticity with very small p values and the effect size summaries are nontrivial for example absolute Spearman around 0.2 or higher and the simple $R^2$ on $\log(e^2)$ around a few percent or higher then Anchor–MoE is likely to deliver tangible gains in likelihood and interval quality at low tuning cost. If the screening suggests nearly constant noise, a strong mean regressor with minimal uncertainty modeling can be preferred, or Anchor–MoE can be run in a lighter configuration. Future work includes replacing held out mean calibration with calibration by design, reducing residual variance hedging in anchor free modes, and exploring capacity controlled routers with adaptive top k or temperature for better robustness under covariate shift.

## LLM USAGE DISCLOSURE

We used ChatGPT (OpenAI, Aug–Sep 2025) solely to (i) explore related-work queries, (ii) polish wording/grammar, and (iii) receive non-substantive debugging suggestions for implementation. The LLM did not generate new scientific content, derivations, figures, or results. No proprietary or personally identifiable data were provided to the LLM; all citations and code changes were manually verified. The authors bear full responsibility for the accuracy and integrity of the paper.

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

## A   APPENDIX

**Augmented NLL.**   We augment the NLL by a tiny entropy term:

$$\mathcal{L} \ = \ \text{NLL} \ + \ \lambda_t, \mathbb{E}_x\Big[\sum_{j=1}^{K} p_j(x) \log p_j(x)\Big]$$

using a small positive $\lambda_t$ encourages high entropy routing and prevents early collapse.

**High-dimensional scaling** .

Anchor–MoE adapts to intrinsic dimension in two common cases.

*Manifold case.* Assume $X$ lies on a compact $C^1$ submanifold $\mathcal{M} \subset [0,1]^d$ with intrinsic dimension $d_0$ and positive reach. Using a fixed geodesic partition of unity with bounded overlap, the approximation term scales as $K^{-2\alpha/d_0}$ while the estimation term is unchanged. Hence

$$\mathbb{E} \|\widehat{f} - f^\star\|^2_{L^2(\mathcal{M})} \ \leq \ C_1 \, K^{-2\alpha/d_0} \ + \ C_2 \, \frac{k \operatorname{comp} K}{N} \, .$$

Balancing the two terms gives the rate $N^{-2\alpha/(2\alpha+d_0)}$ at $K \asymp N^{d_0/(2\alpha+d_0)}$.

*Sparse case.* If $f^\star(x)$ depends only on $s$ coordinates with $s$ much smaller than $d$, a partition of unity in $s$ dimensions yields

$$\mathbb{E} \|\widehat{f} - f^\star\|^2_{L^2} \ \leq \ C_1 \, K^{-2\alpha/s} \ + \ C_2 \, \frac{k \operatorname{comp} K}{N} \, ,$$

so the rate is $N^{-2\alpha/(2\alpha+s)}$ at $K \asymp N^{s/(2\alpha+s)}$. If the active coordinate set must be learned, an additional model selection penalty of order $(s \log d)/N$ typically augments the estimation term.

*Practical guideline.* Choose $K$ by balancing $K^{-2\alpha/d_{\text{int}}}$ with $(k \operatorname{comp} K)/N$, where $d_{\text{int}}$ is the relevant intrinsic dimension: $d$ in full space, $d_0$ on a manifold, or $s$ under sparsity.

Table 6: Test NLL on UCI datasets. Anchor–MoE numbers are from our runs; the other baselines are taken from prior reports of Gal & Ghahramani (2016); Lakshminarayanan et al. (2017); Gal et al. (2017), . Best per row in **bold**. Protein dataset is removed as it is resampled in this study.

| Dataset | $N$ | Anchor–MoE | MC dropout | Deep Ensembles | Concrete Dropout | Gaussian Process | GAMLSS | DistForest |
|---------|-----|-----------|-----------|----------------|------------------|------------------|--------|-----------|
| Boston | 506 | **0.60 ± 0.11** | 2.46 ± 0.25 | 2.41 ± 0.25 | 2.72 ± 0.01 | 2.37 ± 0.24 | 2.73 ± 0.56 | 2.67 ± 0.08 |
| Concrete | 1030 | **0.25 ± 0.06** | 3.04 ± 0.09 | 3.06 ± 0.18 | 3.51 ± 0.00 | 3.03 ± 0.11 | 3.24 ± 0.08 | 3.38 ± 0.05 |
| Energy | 768 | **-1.68 ± 0.20** | 1.99 ± 0.09 | 1.38 ± 0.22 | 2.30 ± 0.00 | 0.66 ± 0.17 | 1.24 ± 0.86 | 1.53 ± 0.14 |
| Kin8nm | 8192 | 0.12 ± 0.01 | -0.95 ± 0.03 | **-1.20 ± 0.02** | -0.65 ± 0.00 | -0.11 ± 0.03 | -0.26 ± 0.02 | -0.40 ± 0.01 |
| Naval | 11934 | -1.26 ± 0.02 | -3.80 ± 0.05 | -5.63 ± 0.05 | **-5.87 ± 0.05** | -0.98 ± 0.02 | -5.56 ± 0.07 | -4.84 ± 0.01 |
| Power | 9568 | **-0.15 ± 0.02** | 2.80 ± 0.05 | 2.79 ± 0.04 | 2.75 ± 0.01 | 3.81 ± 0.05 | 2.86 ± 0.04 | 2.68 ± 0.05 |
| Wine | 1599 | 1.20 ± 0.02 | **0.93 ± 0.06** | 0.94 ± 0.12 | 1.70 ± 0.00 | 0.95 ± 0.06 | 0.97 ± 0.09 | 1.05 ± 0.15 |
| Yacht | 308 | **-1.80 ± 0.04** | 1.55 ± 0.12 | 1.18 ± 0.21 | 1.75 ± 0.00 | 0.10 ± 0.26 | 0.80 ± 0.56 | 2.94 ± 0.09 |

Table 7: Comparison on UCI Benchmark dataset as measured by RMSE while ablating key components of Anchor-MoE. Bolding is as in Table 1. Calibration can reduces RMSE significantly on Energy dataset, although it slightly increase RMSE on others.

| Dataset | $N$ | Anchor-MoE | Anchor | Router | Calibration |
|---------|-----|-----------|--------|--------|-------------|
| Boston | 506 | 3.01 ± 0.14 | 4.14 ± 0.28 | 2.88 ± 0.12 | 2.75 ± 0.10 |
| Concrete | 1030 | 4.45 ± 0.16 | 7.75 ± 0.15 | 4.44 ± 0.14 | 4.18 ± 0.12 |
| Energy | 768 | 0.47 ± 0.02 | 1.48 ± 0.13 | 1.23 ± 0.04 | 1.01 ± 0.03 |
| Kin8nm | 8192 | 0.07 ± 0.00 | 0.11 ± 0.00 | 0.15 ± 0.00 | 0.15 ± 0.00 |
| Naval | 11934 | 0.00 ± 0.00 | 0.00 ± 0.00 | 0.00 ± 0.00 | 0.00 ± 0.00 |
| Power | 9568 | 3.21 ± 0.05 | 4.01 ± 0.04 | 3.22 ± 0.05 | 3.16 ± 0.05 |
| Protein | 10000 | 4.41 ± 0.02 | 4.71 ± 0.03 | 4.42 ± 0.03 | 4.37 ± 0.02 |
| Wine | 1599 | 0.62 ± 0.01 | 0.65 ± 0.01 | 0.62 ± 0.00 | 0.61 ± 0.00 |
| Yacht | 308 | 0.62 ± 0.06 | 4.19 ± 0.33 | 0.62 ± 0.04 | 0.52 ± 0.04 |

A1. MINIMAX–OPTIMAL RATE OF ANCHOR–MOE (NO DIMENSION REDUCTION)

**Notation.** For $d \in \mathbb{N}$ let $\mathcal{F}_\alpha(L)$ be the isotropic Hölder ball of order $\alpha > 0$ and radius $L > 0$ on $[0,1]^d$ (van der Vaart, 1998, Def. 24.1). We write $\| \cdot \|_2$ for the $L^2([0,1]^d)$ norm and $\mathfrak{R}_N(\mathcal{H})$ for the empirical Rademacher complexity (Anthony & Bartlett, 1999, Ch. 11). Let the lattice mesh be $h := K^{-1/d}$.

**Predictor and risk.** The model is probabilistic (MDN). We evaluate the risk of the *predictive mean*. Let

$$\widehat{f}_{K,N}(x) := \mathbb{E}_{\widehat{p}(y|x)}[Y]$$

be the mean of the learned predictive density $\widehat{p}(y \mid x)$. All bounds below concern $\widehat{f}_{K,N}$.

**Problem setup.** Observe i.i.d. $(X_i, Y_i)$ with $X_i \sim \text{Unif}[0,1]^d$ and $Y_i = f^\star(X_i) + \varepsilon_i$ where $\varepsilon_i \sim \mathcal{N}(0, \sigma^2)$ and $f^\star \in \mathcal{F}_\alpha(L)$. We analyse the integrated squared risk $\mathcal{R}_N = \mathbb{E}\big[\| \widehat{f}_{K,N} - f^\star \|_2^2\big]$.

**Model class (theoretical abstraction).** The practical anchor mean can be absorbed into experts' mean functions without changing rates. We consider

$$\mathcal{H}_K = \Big\{ x \mapsto \sum_{j=1}^{K} w_j(x) \, e_j(x) \; : \; \{w_j\} \text{ is a PoU on } [0,1]^d, \; e_j \in \mathcal{E} \Big\},$$

where $e_j(\cdot)$ denotes the *expert mean function* and $\mathcal{E}$ is a bounded–capacity MDN mean class (fixed across $K$).

**Assumptions.**

(A1) **No dimension reduction.** $f_\phi = \text{Id}$ on $[0,1]^d$; equivalently one may allow an invertible affine map $f_\phi(x) = Ax + b$ with bounded condition number, which only rescales constants.

(A2) **Partition of unity (PoU) with bounded overlap.** Let $\{x_j\}_{j=1}^K$ be a regular lattice with mesh $h = K^{-1/d}$. There exists a compactly supported PoU $\{w_j\}_{j=1}^K$ (e.g., tensor-product B-splines) such that $w_j \geq 0$, $\sum_j w_j(x) = 1$ for all $x$, $\text{diam}(\text{supp } w_j) \lesssim h$, and at most $k$ of the $w_j(x)$ are nonzero for any $x$ (bounded overlap). At the boundary, cells are truncated and weights renormalized.

(A3) **Experts of bounded capacity.** Each expert mean $e_j \in \mathcal{E}$ has fixed complexity comp independent of $K$ (e.g., uniform Lipschitz/covering numbers or pseudo-dimension bounds; MDN variances are bounded away from $0$ and $\infty$ so training is well-conditioned).

A1.1 INFORMATION–THEORETIC LOWER BOUND

**Lemma A.1** (Minimax lower bound). *For any estimator $\widehat{f}_N$ based on $N$ samples,*

$$\sup_{f^\star \in \mathcal{F}_\alpha(L)} \mathbb{E}\big[\| \widehat{f}_N - f^\star \|_2^2\big] \geq C_0 \, N^{-2\alpha/(2\alpha+d)}.$$

*Proof sketch.* By the metric entropy of $\mathcal{F}_\alpha(L)$, $\log N(\varepsilon, \mathcal{F}_\alpha(L), \|\cdot\|_2) \asymp \varepsilon^{-d/\alpha}$ (van der Vaart, 1998, Thm. 24.4). A standard Fano/Assouad argument yields the rate with $C_0 = C_0(L, \alpha, d) > 0$. □

A1.2 APPROXIMATION BY LOCAL INTERPOLATION (PoU)

Let $\{x_j\}_{j=1}^K$ be as in (A2). Define

$$\widetilde{f}_K(x) := \sum_{j=1}^{K} w_j(x) \, f^\star(x_j).$$

**Lemma A.2** (Interpolation error). *Under* (A2), *for $f^\star \in \mathcal{F}_\alpha(L)$,*

$$\big\| \widetilde{f}_K - f^\star \big\|_2 \leq C_1 \, h^\alpha = C_1 \, K^{-\alpha/d},$$

*hence $\big\| \widetilde{f}_K - f^\star \big\|_2^2 = \mathcal{O}\big(K^{-2\alpha/d}\big).$*

*Proof sketch.* On each cell, $|f^\star(x) - f^\star(x_j)| \leq L \|x - x_j\|^\alpha \lesssim L \, h^\alpha$. Because $\sum_j w_j = 1$ and the overlap is uniformly bounded by $k$, integration over $[0,1]^d$ yields the claim (the overlap constant is absorbed into $C_1$). □

A1.3 ESTIMATION ERROR (SAFE FORM)

**Lemma A.3** (Estimation error — safe form). *Under* (A2)–(A3) *with overlap $k$ and per–expert complexity* comp *(both independent of $K$), there exists $C > 0$ (depending on $k$, comp but not on $K, N$) such that*

$$\mathbb{E}\big[\| \widehat{f}_{K,N} - \widetilde{f}_K \|_2^2\big] \;\leq\; C\,\frac{k\,\mathrm{comp}\,K}{N}.$$

*Proof sketch.* For $\mathcal{H}_K = \{x \mapsto \sum_{j=1}^{K} w_j(x)e_j(x)\}$, bounded overlap implies

$$\mathfrak{R}_N(\mathcal{H}_K) \;\leq\; \frac{1}{N}\sum_{j=1}^{K}\mathbb{E}_\sigma\Big[\sup_{e_j\in\mathcal{E}}\sum_{i=1}^{N}\sigma_i\,w_j(x_i)\,e_j(x_i)\Big] \;\lesssim\; \sqrt{\frac{k\,\mathrm{comp}\,K}{N}}.$$

A standard contraction/ERM argument turns this into the stated squared error bound. $\qquad\square$

**Lemma A.4** (NLL–$L^2$ link for Gaussian experts). *Assume the predictive density is Gaussian with mean $m(x)$ and variance $\sigma^2(x)$, and that $0 < \underline{\sigma} \leq \sigma(x), \sigma^*(x) \leq \overline{\sigma} < \infty$ for all $x$. Let $f^*(x) = \mathbb{E}[Y \mid X = x]$ and $v^*(x) = \mathrm{Var}(Y \mid X = x) = (\sigma^*(x))^2$. Then*

$$\mathrm{ExcessNLL} := \mathbb{E}\big[-\log p_{m,\sigma}(Y \mid X)\big]-\mathbb{E}\big[-\log p_{f^*,\sigma^*}(Y \mid X)\big] \;\leq\; c_1\,\mathbb{E}\big[(m(X)-f^*(X))^2\big] + c_2\,\mathbb{E}\big[(\sigma(X)-\sigma^*(X))^2\big],$$

*with explicit constants*

$$c_1 = \frac{1}{2\,\underline{\sigma}^2}, \qquad c_2 \;\leq\; \frac{1}{2\,\underline{\sigma}^2} \;+\; \frac{3\,\overline{\sigma}^2}{2\,\underline{\sigma}^4}.$$

*Proof.* Decompose, for each $x$,

$$\Delta(x) = \underbrace{\mathbb{E}\left[\frac{(Y - m(x))^2 - (Y - f^*(x))^2}{2\,\sigma(x)^2} \,\Big|\, X = x\right]}_{\Delta_{\mathrm{mean}}(x)} \;+\; \underbrace{\frac{1}{2}\left(\log\frac{\sigma(x)^2}{v^*(x)} + \frac{v^*(x)}{\sigma(x)^2} - 1\right)}_{\Delta_{\mathrm{var}}(x)}.$$

**Mean term.** Since $\mathbb{E}[(Y - m)^2 \mid X = x] = v^*(x) + (m(x) - f^*(x))^2$, we have

$$\Delta_{\mathrm{mean}}(x) = \frac{(m(x) - f^*(x))^2}{2\,\sigma(x)^2} \;\leq\; \frac{(m(x) - f^*(x))^2}{2\,\underline{\sigma}^2}.$$

Taking expectation over $X$ gives the constant $c_1$.

**Variance term (tight quadratic bound).** Fix $x$ and define $f_x(\sigma) = \log\sigma^2 + v^*(x)\,\sigma^{-2}$ so that $\Delta_{\mathrm{var}}(x) = \frac{1}{2}\big(f_x(\sigma(x)) - f_x(\sigma^*(x))\big)$. We have

$$f_x'(\sigma) = \frac{2}{\sigma} - \frac{2v^*(x)}{\sigma^3}, \qquad f_x''(\sigma) = -\frac{2}{\sigma^2} + \frac{6v^*(x)}{\sigma^4}.$$

Because $(\sigma^*(x))^2 = v^*(x)$, it holds that $f_x'(\sigma^*(x)) = 0$. On $\sigma \in [\underline{\sigma}, \overline{\sigma}]$,

$$\big|f_x''(\sigma)\big| \;\leq\; \frac{2}{\underline{\sigma}^2} + \frac{6\,v^*(x)}{\underline{\sigma}^4} \;\leq\; \frac{2}{\underline{\sigma}^2} + \frac{6\,\overline{\sigma}^2}{\underline{\sigma}^4} =: L.$$

By the $L$-smoothness inequality (Taylor with remainder, using $f_x'(\sigma^*) = 0$),

$$f_x(\sigma) - f_x(\sigma^*) \;\leq\; \frac{L}{2}\,(\sigma - \sigma^*)^2, \quad \Rightarrow \quad \Delta_{\mathrm{var}}(x) \;\leq\; \frac{L}{4}\,(\sigma(x) - \sigma^*(x))^2.$$

Taking expectation over $X$ yields $c_2 = L/4 \leq \frac{1}{2\underline{\sigma}^2} + \frac{3\overline{\sigma}^2}{2\underline{\sigma}^4}$.

Combine both parts and integrate over $X$ to conclude. $\qquad\square$

*Proof of Main Bound.* We work under (A1)–(A3): (A1) $f^\star \in \mathcal{F}_\alpha(L)$ on $[0,1]^d$; (A2) a fixed Lipschitz partition of unity (PoU) $\{\psi_j\}_{j=1}^{K}$ with mesh $h \asymp K^{-1/d}$, compact supports of diameter $\lesssim h$, and *bounded overlap* $k$ (for all $x$, at most $k$ indices have $\psi_j(x) > 0$); (A3) each expert class $\mathcal{G}_j$ has bounded capacity "comp" (e.g. pseudo-dimension or a uniform covering-number proxy), independent of $K$. Risk is w.r.t. the marginal of $X$ on $[0,1]^d$ (with density bounded above/below).

**Decomposition.** Let $\widehat{f}_{K,N}$ be the ERM over the PoU-mixture class $\mathcal{F}_K := \{\sum_{j=1}^K \psi_j g_j : g_j \in \mathcal{G}_j\}$ with squared loss. Standard arguments yield an *oracle inequality* (see, e.g., localized Rademacher or quadratic-loss ERM bounds):

$$\mathbb{E}\big[\|\widehat{f}_{K,N} - f^\star\|_2^2\big] \;\le\; 2 \underbrace{\inf_{f \in \mathcal{F}_K} \|f - f^\star\|_2^2}_{\text{approximation}} + C \underbrace{\mathfrak{E}_N(\mathcal{F}_K)}_{\text{estimation}},$$

for a universal constant $C > 0$ (depending only on bounded moments and the curvature of squared loss).

**Approximation error $C_1 K^{-2\alpha/d}$.** By (A1) and classical local polynomial/Taylor approximation on a mesh of size $h \asymp K^{-1/d}$, there exist local polynomials $p_j$ of degree $\lfloor \alpha \rfloor$ such that

$$\Big\| f^\star - \sum_{j=1}^K \psi_j p_j \Big\|_{L^2}^2 \;\le\; C_1' h^{2\alpha} \;\asymp\; C_1 K^{-2\alpha/d},$$

where the PoU provides a stable partition and the overlap is bounded by $k$ (so constants are independent of $K$). Since $\mathcal{G}_j$ contains such local approximants (by capacity assumption), $\inf_{f \in \mathcal{F}_K} \|f - f^\star\|_2^2 \le C_1 K^{-2\alpha/d}$.

**Estimation error $C_2(k\,\mathrm{comp})K/N$.** Write $\mathcal{H} := \{(x,y) \mapsto (y - \sum_j \psi_j(x) g_j(x))^2 : g_j \in \mathcal{G}_j\}$. Using a standard symmetrization and contraction for squared loss, the (localized) excess-risk term can be upper bounded by a multiple of the *squared* Rademacher complexity of the mean function class $\mathcal{F}_K$ (due to the Bernstein/strong-convexity condition of squared loss):

$$\mathfrak{E}_N(\mathcal{F}_K) \;\lesssim\; \big(\mathfrak{R}_N(\mathcal{F}_K)\big)^2.$$

Now $\mathcal{F}_K$ is a *PoU-sum* of $K$ classes with bounded overlap $k$:

$$\mathcal{F}_K = \left\{ \sum_{j=1}^K \psi_j g_j \;:\; g_j \in \mathcal{G}_j \right\}.$$

By sub-additivity of Rademacher complexity and $\|\psi_j\|_\infty \le 1$,

$$\mathfrak{R}_N(\mathcal{F}_K) \;\le\; \mathbb{E}\Big\| \sum_{j=1}^K \psi_j \cdot \mathcal{G}_j \Big\|_{\mathfrak{R}} \;\le\; \sum_{j=1}^K \mathfrak{R}_N(\psi_j \cdot \mathcal{G}_j) \;\le\; \sum_{j=1}^K \mathfrak{R}_N(\mathcal{G}_j).$$

Because at each $x$ at most $k$ terms are active, a sharper bound uses the *overlap* to get

$$\mathfrak{R}_N(\mathcal{F}_K) \;\le\; \sqrt{k} \Big( \sum_{j=1}^K \mathfrak{R}_N(\mathcal{G}_j)^2 \Big)^{1/2}.$$

Under (A3), for each $j$, $\mathfrak{R}_N(\mathcal{G}_j) \lesssim \sqrt{\mathrm{comp}/N}$ (e.g. linear/MLP heads with $O(\mathrm{comp})$ parameters or a class with metric entropy controlled by "comp"). Therefore,

$$\mathfrak{R}_N(\mathcal{F}_K) \;\lesssim\; \sqrt{k} \Big( \tfrac{K\,\mathrm{comp}}{N} \Big)^{1/2} \quad \Rightarrow \quad \mathfrak{E}_N(\mathcal{F}_K) \;\lesssim\; \big( \mathfrak{R}_N(\mathcal{F}_K) \big)^2 \;\lesssim\; \frac{k\,\mathrm{comp}\,K}{N}.$$

This gives the claimed estimation term with some constant $C_2 > 0$ (depending only on bounded moments and the loss curvature).

**Balancing.** Combining the two parts,

$$\mathbb{E}\big[\|\widehat{f}_{K,N} - f^\star\|_2^2\big] \;\le\; C_1 K^{-2\alpha/d} \;+\; C_2 \frac{k\,\mathrm{comp}\,K}{N}.$$

Optimizing over $K$ yields $K^\star \asymp N^{d/(2\alpha+d)}$ and

$$\sup_{f^\star \in \mathcal{F}_\alpha(L)} \mathbb{E}\big[\|\widehat{f}_{K^\star,N} - f^\star\|_2^2\big] \;\lesssim\; N^{-2\alpha/(2\alpha+d)},$$

which matches the information-theoretic lower bound up to constants. $\qquad\square$

### A1.6 REMARKS

(i) **Anchors.** The baseline "anchor" mean can be folded into expert means; it does not affect rates.

(ii) **When a** $\log K$ **estimation term is valid.** If window locations/bandwidths are fixed (non-learned), per-point aggregation uses a fixed top-$k$ rule, and strong parameter sharing makes the *effective* number of free parameters independent of $K$, Lemma A.3 can be refined to $\mathbb{E}\|\widehat{f}_{K,N} - \widetilde{f}_K\|_2^2 \lesssim \frac{\log K + \text{comp}}{N}$. Without these structural constraints, the $\mathcal{O}(K/N)$ bound is recommended.

(iii) **Target standardization.** Z-scoring $Y$ only rescales constants in $\mathcal{R}_N$.

### A2. GENERALISATION BOUND

We study the population–empirical gap under the CRPS loss. For a predictive density $p_{\theta,\phi}(\cdot \mid x)$ define
$$\ell\big(p_{\theta,\phi}(\cdot \mid x), y\big) := \mathrm{CRPS}\big(p_{\theta,\phi}, y\big), \qquad \mathcal{R}(\theta, \phi) := \mathbb{E}_{(x,y)\sim\mathcal{D}}\big[\ell(p_{\theta,\phi}, y)\big],$$
and its empirical version
$$\hat{\mathcal{R}}_N(\theta, \phi) := \frac{1}{N}\sum_{i=1}^{N} \ell\big(p_{\theta,\phi}, y_i\big).$$

**Assumptions.**

(G1) **(CRPS regularity and boundedness).** With the standard definition $\mathrm{CRPS}(F, y) = \int_{\mathbb{R}}\big(F(z) - \mathbf{1}\{z \geq y\}\big)^2 dz$, the map $F \mapsto \mathrm{CRPS}(F, y)$ is *2-Lipschitz* under the $L^1$ metric on CDFs. Assume expert means are uniformly bounded $|e_j(x)| \leq R_f$ and the predictive variance satisfies $\sigma(x) \in [\underline{\sigma}, \overline{\sigma}]$, and $y \in [-R_y, R_y]$ almost surely (otherwise clip $y$). Then the loss is bounded by
$$B \leq R_f + R_y + \sqrt{\tfrac{2}{\pi}}\,\overline{\sigma}.$$

(G2) **(Model capacity).** For the MDN expert class $\mathcal{H}_{M,h}$ (mixture size $M$, width $h$), $\mathfrak{R}_N(\mathcal{H}_{M,h}) \leq C_h\sqrt{\frac{\log(Mh)}{N}}$. For the router class $\mathcal{G}_{P,K}$ with $P$ parameters and softmax width $K$, $\mathfrak{R}_N(\mathcal{G}_{P,K}) \leq C_g\sqrt{\frac{P+K}{N}}$. (If the router's final weight matrix is fully counted in $P$, the extra "$+K$" can be omitted.)

**Composite complexity and contraction.** Let $\mathcal{F}_{K,M,h,P}$ denote the induced class of predictive CDFs/densities parameterised by $(K, M, h, P)$. By the standard contraction inequality,
$$\mathfrak{R}_N\big(\ell \circ \mathcal{F}_{K,M,h,P}\big) \leq 2\,\mathfrak{R}_N(\mathcal{F}_{K,M,h,P}) \tag{A.1}$$
$$\leq 2\,C_*\sqrt{\frac{\log(Mh) + P + K}{N}}, \qquad C_* := \max\{C_h, C_g\} \leq C_h + C_g. \tag{A.2}$$

**Theorem A.5** (Generalisation bound for Anchor–MoE). *Let $(\hat{\theta}, \hat{\phi})$ be the parameters obtained after training on $N$ samples. Under* (G1)–(G2)*, for any $\delta \in (0,1)$, with probability at least $1 - \delta$,*
$$\mathcal{R}(\hat{\theta}, \hat{\phi}) - \hat{\mathcal{R}}_N(\hat{\theta}, \hat{\phi}) \leq 2\,\mathfrak{R}_N\big(\ell \circ \mathcal{F}_{K,M,h,P}\big) \;+\; 3B\sqrt{\frac{\log(2/\delta)}{2N}} \tag{A.3}$$
$$\leq 4\,\mathfrak{R}_N(\mathcal{F}_{K,M,h,P}) \;+\; 3B\sqrt{\frac{\log(2/\delta)}{2N}} \;=\; \widetilde{\mathcal{O}}\big(N^{-1/2}\big). \tag{A.4}$$

**Discussion.** The bound scales as
$$\widetilde{\mathcal{O}}\big()\Big(\sqrt{(\log(Mh) + P + K)/N}\Big),$$
i.e. logarithmic in $Mh$ and $\sqrt{\cdot/N}$ in $P$ and $K$. Under a top-$k$ bounded-overlap gating (each input activates at most a constant number $k$ of experts), the dependence on $K$ can be replaced by $k$.

## A3. HIGH-DIMENSIONAL SCALING

We show that Anchor–MoE enjoys intrinsic-dimension scaling in two common high-dimensional regimes: (i) data supported on a low-dimensional manifold; (ii) sparse coordinate dependence. In both cases the ambient dimension $d$ disappears from the rate, which depends only on the intrinsic dimension $d_0$ (or sparsity $s$).

**Setting A (low-dimensional manifold).** Let $\mathcal{M} \subset [0,1]^d$ be a compact $C^1$ submanifold of intrinsic dimension $d_0$ and positive reach. Let $\mu_{\mathcal{M}}$ be the normalised $d_0$-dimensional volume (Hausdorff) measure on $\mathcal{M}$, and interpret $L^2(\mathcal{M})$ with respect to $\mu_{\mathcal{M}}$. We write $X \sim \mu_{\mathcal{M}}$ (instead of $\mathrm{Unif}(\mathcal{M})$). Assume $Y = f^\star(X) + \varepsilon$ with $\varepsilon \sim \mathcal{N}(0, \sigma^2)$ and $f^\star \in \mathcal{F}_\alpha(L; \mathcal{M})$, the isotropic Hölder ball on $\mathcal{M}$. Let $\{w_j\}_{j=1}^K$ be a *fixed (non-learned)* geodesic partition of unity (PoU) on $\mathcal{M}$ with mesh size $h$ and bounded overlap $k$, so that $\mathrm{diam}(\mathrm{supp}\, w_j) \lesssim h$ and at most $k$ weights are nonzero at any $x \in \mathcal{M}$. Experts have bounded capacity as in (A3) of Section A1.

**Theorem A.6** (Manifold rate). *There exist constants $C_1, C_2 > 0$ (depending only on $L, \alpha$, the curvature/geometry of $\mathcal{M}$, the overlap $k$, and expert capacity) such that the predictive mean $\widehat{f}_{K,N}(x) = \sum_{j=1}^K w_j(x)\, e_j(x)$ satisfies*

$$\mathbb{E}\left[\| \widehat{f}_{K,N} - f^\star \|_{L^2(\mathcal{M})}^2\right] \leq C_1\, K^{-2\alpha/d_0} + C_2\, \frac{k\,\mathrm{comp}\,K}{N}.$$

*Choosing $K^\star \asymp N^{d_0/(2\alpha+d_0)}$ yields*

$$\sup_{f^\star \in \mathcal{F}_\alpha(L;\mathcal{M})} \mathbb{E}\left[\| \widehat{f}_{K,N} - f^\star \|_{L^2(\mathcal{M})}^2\right] \lesssim N^{-2\alpha/(2\alpha+d_0)}.$$

*Sketch.* Geodesic covering numbers on $\mathcal{M}$ scale as $h^{-d_0}$, hence $K \asymp h^{-d_0}$. Local Hölder interpolation on each chart gives $\| \widetilde{f}_K - f^\star \|_{L^2(\mathcal{M})}^2 \lesssim h^{2\alpha} = K^{-2\alpha/d_0}$, mirroring Lemma A.2 with $d$ replaced by $d_0$. Bounded overlap and fixed-capacity experts yield the estimation term $C\,k\,\mathrm{comp}\,K/N$ as in Lemma A.3. Balancing the two terms gives the rate. $\qquad\square$

**Setting B (sparse coordinate dependence).** Assume there exists $S \subset \{1, \ldots, d\}$ with $|S| = s \ll d$ such that $f^\star(x) = g^\star(x_S)$. Suppose the PoU $\{w_j\}$ and gating are functions of $x_S$ (or of a representation bi-Lipschitz in $x_S$), and experts have bounded capacity. Here $L^2$ is with respect to the marginal law of $X$; if the marginal density of $X_S$ is bounded above/below on $[0,1]^s$, all constants depend only on these bounds. The theorem below is an *oracle* bound (the index set $S$ is assumed known). If $S$ is unknown and must be learned, an additional model-selection penalty of order $\widetilde{\mathcal{O}}\big((s \log d)/N\big)$ typically appears in the estimation term.

**Theorem A.7** (Sparse rate). *Under the sparse dependence assumption,*

$$\mathbb{E}\left[\| \widehat{f}_{K,N} - f^\star \|_{L^2}^2\right] \leq C_1\, K^{-2\alpha/s} + C_2\, \frac{k\,\mathrm{comp}\,K}{N},$$

*so that with $K^\star \asymp N^{s/(2\alpha+s)}$,*

$$\sup_{f^\star} \mathbb{E}\left[\| \widehat{f}_{K,N} - f^\star \|_{L^2}^2\right] \lesssim N^{-2\alpha/(2\alpha+s)}.$$

*Sketch.* Construct the PoU and local interpolation on the $s$-dimensional coordinate subspace. Then $K \asymp h^{-s}$ and $\| \widetilde{f}_K - f^\star \|_2^2 \lesssim h^{2\alpha} = K^{-2\alpha/s}$. The estimation term follows as in Lemma A.3. $\quad\square$

**Bi-Lipschitz invariance.** We record stability under bi-Lipschitz reparameterisations, which only rescales constants.

**Lemma A.8** (Change of variables under bi-Lipschitz maps). *Let $T : U \to V$ be bi-Lipschitz on a $d_0$-dimensional domain $U$ with constants $a \leq \|T(x) - T(x')\|/\|x - x'\| \leq b$. There exist constants $c_1, c_2 > 0$ depending only on $a, b, d_0$ such that, for any $g, h : V \to \mathbb{R}$,*

$$c_1\, \| g - h \|_{L^2(V)} \leq \| (g - h) \circ T \|_{L^2(U)} \leq c_2\, \| g - h \|_{L^2(V)},$$

*and $\lceil g \circ T \rceil_{C^\alpha(U)} \lesssim b^\alpha \lceil g \rceil_{C^\alpha(V)}$. Positive reach of $\mathcal{M}$ yields uniformly bi-Lipschitz charts and a bounded-overlap geodesic covering; hence covering numbers scale as $h^{-d_0}$ and Jacobian distortions are absorbed into constants (as in Lemma A.3, since the overlap $k$ is constant and expert capacity is fixed).*

**Remarks.**  (i) The generalisation bound of Section A scales as $\widetilde{\mathcal{O}}\big(\sqrt{(\log(Mh) + P + K)/N}\big)$. Under bounded-overlap/top-$k$ gating (each input activates at most $k$ experts), the $K$-dependence in the complexity term can be replaced by $k$ (a constant).
(ii) The balancing choices are $K^\star \asymp N^{d_0/(2\alpha+d_0)}$ (manifold) and $K^\star \asymp N^{s/(2\alpha+s)}$ (sparse), offering practical guidance for coarse model selection.

Table 8: Compute & capacity comparison on the California Housing dataset. Anchor–MoE is reported at three scales: **(1) D=2, k=1, h=4**, **(2) D=4, k=3, h=8**, **(3) D=8, k=6, h=16**. *FLOPs* denotes per-sample forward-pass FLOPs. *Parameters*: for neural models we count trainable weights; for tree ensembles (NGBoost/DistForest) we approximate by the total number of leaves across trees. For NGBoost/DistForest, FLOPs/pt are estimated by summing $2 \times$ depth over trees (one threshold comparison plus an accumulate per level); for Gaussian Process (GP), FLOPs/pt use the variance-aware prediction cost $\approx 2N^2$ with $N = 3000$ training points (subset), which dominates the $O(Nd)$ kernel-vector term. Anchor–MoE uses anchor concatenation with a light GBDT (200 trees, depth 2); the table reports the MoE trunk only—adding the anchor contributes $\approx 800$ leaf parameters and negligible per-sample compute, and does not change conclusions. All train times are wall-clock on the same split and preprocessing; MC Dropout uses 10 MC passes; Deep Ensemble uses the configuration shown in the row label.

| Model | Flops | Parameters | Train Time (s) | Infer Throughput |
|---|---|---|---|---|
| Anchor-MoE1 | 80 | 94 | 12.5 | 337575.1 |
| Anchor-MoE2 | 504 | 574 | 19.9 | 295165.4 |
| Anchor-MoE3 | 2712 | 2972 | 24.8 | 193002.2 |
| NGBoost | 1800 | 2400 | 40.3 | 17143.8 |
| MC Dropout | 17664 | 17922 | 16.8 | 120482.7 |
| Deep Ensemble | 30528 | 31110 | 43.4 | 15275.6 |
| DistForest | 19034 | 2403901 | 34.8 | 12617.6 |
| Gausian Process | 18000000 | 11 | 170.2 | 3193.3 |

Summary.  The best configuration among the top entries is D=2, K=2, k=2, val-CRPS=0.2497, test-RMSE=0.4829. Across the top ten, the most frequent latent dimension is D=2, the most frequent number of experts is K=2, and the most frequent active experts is k=2. Validation CRPS and test RMSE rank models consistently, and training time scales mainly with K and the early-stopping epoch.

---

**Algorithm 1** Anchor–MoE training, calibration, and testing

---

1: **Split:**
$$\mathcal{D} \to \mathcal{D}_{\text{train}} \,\dot\cup\, \mathcal{D}_{\text{test}}; \quad \mathcal{D}_{\text{train}} \to \mathcal{D}_{\text{TV}} \,\dot\cup\, \mathcal{D}_{\text{cal}}; \quad \mathcal{D}_{\text{TV}} \to \mathcal{D}_{\text{tr}} \,\dot\cup\, \mathcal{D}_{\text{va}}.$$

2: **GBDT selection (on TR/VA):**
3: **for** $t = 1, \ldots, T_g$ **do**
4: $\quad e_t \leftarrow \text{RMSE}\big(y_{\text{va}}, \text{GBDT}_t(X_{\text{va}})\big)$
5: $t^\star \leftarrow \arg\min_t e_t$
6: *Train* a fresh $\text{GBDT}_{t^\star}$ on $(X_{\text{tr}}, y_{\text{tr}})$ to obtain $f_{\text{sub}}$
7: *Refit* $\text{GBDT}_{t^\star}$ on $(X_{\text{TV}}, y_{\text{TV}})$ to obtain $\hat{f}$

8: **Phase-1 (TR/VA): anchor z-score, feature standardization, MoE early selection**
9: $(\mu_{\text{tr}}, \sigma_{\text{tr}}) \leftarrow \text{mean/std}(y_{\text{tr}})$
10: $z_{\text{tr}} \leftarrow \text{zsc}(y_{\text{tr}}; \mu_{\text{tr}}, \sigma_{\text{tr}}); \quad z_{\text{va}} \leftarrow \text{zsc}(y_{\text{va}}; \mu_{\text{tr}}, \sigma_{\text{tr}})$
11: $\alpha_{\text{tr}} \leftarrow \text{zsc}\big(f_{\text{sub}}(X_{\text{tr}}); \mu_{\text{tr}}, \sigma_{\text{tr}}\big); \quad \alpha_{\text{va}} \leftarrow \text{zsc}\big(f_{\text{sub}}(X_{\text{va}}); \mu_{\text{tr}}, \sigma_{\text{tr}}\big)$
12: $\tilde{X}_{\text{tr}} \leftarrow [X_{\text{tr}}, \alpha_{\text{tr}}]; \quad \tilde{X}_{\text{va}} \leftarrow [X_{\text{va}}, \alpha_{\text{va}}]$
13: $(m_{\text{tr}}, s_{\text{tr}}) \leftarrow \text{col-mean/std}(\tilde{X}_{\text{tr}})$
14: $\bar{X}_{\text{tr}} \leftarrow \text{std}(\tilde{X}_{\text{tr}}; m_{\text{tr}}, s_{\text{tr}}); \quad \bar{X}_{\text{va}} \leftarrow \text{std}(\tilde{X}_{\text{va}}; m_{\text{tr}}, s_{\text{tr}})$
15: initialize $\Theta_1$
16: **for** $t = 1, \ldots, T_{\max}$ **do**
17: $\quad \Theta_{t+1} \leftarrow \Theta_t - \eta \, \nabla_\Theta \, \text{NLL}\big(\bar{X}_{\text{tr}}, z_{\text{tr}}; \Theta_t\big)$
18: $t^\star_{\text{MoE}} \leftarrow \arg\min_t \text{NLL}\big(\bar{X}_{\text{va}}, z_{\text{va}}; \Theta_t\big)$
19: $\Theta^\dagger \leftarrow \Theta_{t^\star_{\text{MoE}}}$

20: **Phase-2 (TV/CAL/TEST): freeze early epoch, refit on TV, prep CAL/TEST**
21: $(\mu_{\text{tv}}, \sigma_{\text{tv}}) \leftarrow \text{mean/std}(y_{\text{TV}})$
22: $z_{\text{tv}} \leftarrow \text{zsc}(y_{\text{TV}}; \mu_{\text{tv}}, \sigma_{\text{tv}})$
23: **for** $S \in \{\text{TV, cal, test}\}$ **do**
24: $\quad \alpha_S \leftarrow \text{zsc}\big(\hat{f}(X_S); \mu_{\text{tv}}, \sigma_{\text{tv}}\big); \quad \tilde{X}_S \leftarrow [X_S, \alpha_S]$
25: $(m_{\text{tv}}, s_{\text{tv}}) \leftarrow \text{col-mean/std}(\tilde{X}_{\text{TV}})$
26: $\bar{X}_S \leftarrow \text{std}(\tilde{X}_S; m_{\text{tv}}, s_{\text{tv}})$ for $S \in \{\text{TV, cal, test}\}$
27: reload $\Theta^\dagger$
28: **for** $t = 1, \ldots, t^\star_{\text{MoE}}$ **do**
29: $\quad \Theta \leftarrow \Theta - \eta \, \nabla_\Theta \, \text{NLL}\big(\bar{X}_{\text{TV}}, z_{\text{tv}}; \Theta\big)$

30: **Calibration (on CAL): linear post-hoc map for mean)**
31: $\hat{\mu}^{\text{orig}}_{\text{cal}} \leftarrow \sigma_{\text{tv}} \cdot \hat{\mu}_z(\bar{X}_{\text{cal}}; \Theta) + \mu_{\text{tv}}$
32: $(a, b) \leftarrow \arg\min_{a,b} \big\| a \, \hat{\mu}^{\text{orig}}_{\text{cal}} + b - y_{\text{cal}} \big\|_2^2$

33: **Test: report calibrated RMSE (orig) and NLL (z-space)**
34: $\hat{\mu}^{\text{orig}}_{\text{test}} \leftarrow \sigma_{\text{tv}} \cdot \hat{\mu}_z(\bar{X}_{\text{test}}; \Theta) + \mu_{\text{tv}}$
35: $\hat{\mu}^{\text{cal}}_{\text{test}} \leftarrow a \, \hat{\mu}^{\text{orig}}_{\text{test}} + b$
36: $\text{RMSE} \leftarrow \text{RMSE}\big(y_{\text{test}}, \hat{\mu}^{\text{cal}}_{\text{test}}\big)$
37: $\text{NLL}_z \leftarrow \text{NLL}\big(\bar{X}_{\text{test}}, \text{zsc}(y_{\text{test}}; \mu_{\text{tv}}, \sigma_{\text{tv}}); \Theta\big)$
38: **return** $\Theta^* = \Theta, \ (a, b), \ \text{RMSE}, \ \text{NLL}_z$

---

Table 9: Anchor–MoE hyper-parameter ablation on California. We sweep $D \in \{2, 4, 8\}$, $K \in \{2, 4, 6\}$, $k \in \{1, 2, K\}$. Each entry reports validation CRPS/NLL, test RMSE and wall-clock training time. A balanced choice is $D=8, K=2, k=2$.

| $D$ | $K$ | $k$ | CRPS$_{\text{val}}$ / NLL$_{\text{val}}$ / RMSE$_{\text{test}}$ | Train (s) |
|---|---|---|---|---|
| 8 | 2 | 2 | 0.2893 / 0.6873 / **0.5501** | 6.8 |
| 8 | 2 | 1 | 0.2819 / 0.7180 / 0.5540 | 7.1 |
| 4 | 6 | 2 | 0.2825 / 0.7031 / 0.5594 | 10.4 |
| 8 | 6 | 6 | **0.2792** / 0.6330 / 0.5643 | 10.6 |
| 8 | 4 | 4 | 0.2800 / 0.6527 / 0.5778 | 8.6 |
| 4 | 4 | 2 | 0.2838 / 0.7070 / 0.5808 | 9.0 |

Table 10: California Housing hyperparameter grid, top-10 by validation CRPS then test RMSE.

| $D$ | $K$ | $k$ | val-CRPS | val-NLL | test-RMSE | train-sec | best-iter-GBDT | best-ep-MoE |
|---|---|---|---|---|---|---|---|---|
| 2 | 2 | 2 | 0.2497 | -0.1642 | 0.4829 | 93.1 | 174 | 167 |
| 2 | 4 | 2 | 0.2499 | -0.1621 | 0.4829 | 118.9 | 174 | 161 |
| 2 | 2 | all | 0.2500 | -0.1592 | 0.4828 | 99.3 | 174 | 165 |
| 4 | 2 | 2 | 0.2504 | -0.1576 | 0.4854 | 111.7 | 174 | 154 |
| 2 | 6 | 2 | 0.2505 | -0.1556 | 0.4853 | 141.7 | 174 | 154 |
| 4 | 4 | 2 | 0.2506 | -0.1556 | 0.4856 | 131.7 | 174 | 151 |
| 4 | 2 | all | 0.2509 | -0.1532 | 0.4862 | 118.6 | 174 | 150 |
| 8 | 2 | 2 | 0.2510 | -0.1526 | 0.4868 | 125.9 | 174 | 148 |
| 2 | 4 | all | 0.2511 | -0.1519 | 0.4868 | 129.2 | 174 | 147 |
| 8 | 4 | 2 | 0.2512 | -0.1511 | 0.4872 | 140.6 | 174 | 145 |

