# OpenReview forum: "Anchor–MoE: A Mean-Anchored Mixture of Experts for Probabilistic Regression"
_ICLR.cc/2026/Conference — Submitted to ICLR 2026_

### Official Review · Reviewer_JopS · 2025-10-29

**Soundness:** 3
**Presentation:** 2
**Contribution:** 3
**Rating:** 6
**Confidence:** 2

**Summary:**

This paper describes a mixture of experts regression model (Anchor-MoE) for making probabilistic and point predictions. The method incorporates an optional anchor, a (point-estimate) regressor that can be freely chosen;   a gradient boosted decision trees model is used as anchor for the purpose of evaluations.  The experts are lightweight mixture density networks and a soft top-k router  operates on a latent space representation of covariate  inputs concatenated with the anchor prediction to produce k non-zero weights for sparsely combining experts. Theoretical analysis shows (under specified assumptions) that  minimax-optimal convergence rate for MSE can be achieved also covering the cases in which data is on a manifold or has sparse dependency on covariate components.  Bounds relating to CRPS and NLL generalization are also derived.  These theoretical results are used to guide some of the parameter choices in implementation. Ablation studies are provided for useful insights , for example to help understand the benefit of using an anchor.  Anchor-MoE is shown to perform competitively with other probabilistic regression methods, in particular NGboost, across standard UCI-benchmark datasets.

**Strengths:**

The paper presents significant new material: Demonstrating competitive performance of an MoE approach against other respected  mainstream probabilistic regression methodologies, in particular NGBoost, seems like an important result of interest to the ML community;  I have been unable to find other MoE performance comparisons of this nature.  The paper is further strengthened by derivations of bounds for MSE rates of convergence  and  bounds for CRPS and NLL generalization as well as capturing the high dimensional cases of data on manifolds and sparse dependency on covariate coordinates. I have again been unable to find such derivations/result in literature relating to similar MoE approaches.

**Weaknesses:**

The paper seems more inaccessible than it needs to be for non-domain experts who would benefit from fuller explanations and definitions in places. The reasoning underlying the construction of some of the  Anchor-MoE components could be better motivated for the non-MoE-specialist reader (through use of references and/or commentary) . Improvements in these areas would open up the paper to a wider readership. Convergence and related proofs in the appendix are very concise, taking for granted a significant amount of prior knowledge on the part of the reader and making it challenging and time consuming to gain confidence in their validity. Addressing these points would further strengthen the paper.

**Questions:**

1. How is the synthetic 1-d dataset constructed - for example are the oscillations in the point predictions recovered by anchor+MoE   in figure 4 also present in the true synthetic data and if not can you comment on why they might arise. It would be helpful to include the true expected values of the synthetic data on the plots

2. It’s unclear in the ablation analysis of table 2 precisely what has been added, changed or removed in each of the columns. Could this be expanded upon (both in the rebuttal response and  in the paper).  What is the message I should come away with from figure 3 ?  An expanded legend-text and/or main-text commentary of what the reader is meant to deduce from this figure would be helpful.

3. Do the covariate dimensions or other characteristics of the baseline datasets reveal anything interesting about where Anchor-MoE performs well or not so well in comparison to NGBoost and/or the other baseline methods?

4. Can you comment on how reasonable the assumptions A1-A3 and G1-G2 are in relation to  Anchor-MoE.

(Some  minor editorial points I detected: Several cross references are incorrectly made to section 3 instead of section 2. No bolding is provided in table 4.  The sentence at the end of the second paragraph of the introduction is missing an ending. In Lemma A1 shouldn’t the inequality be $\le$ rather than $\ge$?)

---

> ### Author Response · Authors · 2025-11-19
> **Reply ro reviewer JopS**
>
> Q1.
>
> The toy data is generated by a fixed ground–truth function(see details in the file 'generateFigures.ipynb') and does not include the oscillation.The visible oscillations in Fig4 are produced by the learned Anchor--MoE (due to local experts with soft top-$k$ routing and finite capacity). We will overlay the true mean and variance on the plots for clarity.
>
> Q2.
>
> For the ablation experiment, each column exactly one component while keeping all others unchanged . We will add a brief description in the caption.
>
> Q3.
>
> Yes. Across the 9 UCI tasks we observe consistent patterns linking dataset
> characteristics to the relative gap between Anchor--MoE and NGBoost.
>
> Datasets with pronounced heteroscedasticity behavior (e.g.,
> energy and Protein) yield the largest NLL gains for
> Anchor--MoE.
>
> On tasks that are close to homoscedastic and smoothly predictable (e.g.,
> Naval), NGBoost already achieves
> very low error; Anchor--MoE typically matches but does not strongly surpass it.
>
> Q4.
>
> For A1, Our rate proof fixes $f_\phi=\mathrm{Id}$ to avoid mixing approximation and estimation effects. In the implementation we optionally use a shallow linear map with LayerNorm before routing. This map is near bi-Lipschitz on the support of $X$, so it only rescales constants (cf.\ the bi-Lipschitz remark in App.~A3) and leaves the $N^{-2\alpha/(2\alpha+d)}$ rate unchanged. Empirically, ablating the projection has negligible effect on the trends reported.
>
> For A2. Our learnable metric--window plus top-$k$ gating enforces that at most $k$ experts are active per $x$, which is exactly the ``bounded overlap'' surrogate used in the analysis. The window outputs are normalised (with a small $\varepsilon$ smoothing), so they behave like an approximate PoU; window scales are clamped to a compact range, preventing pathological global support. Thus A2 is satisfied up to standard constants.
>
> For A3. Each expert is a small MDN with fixed width and a fixed number of mixture components; means and log-scales are regularised and variances are clamped to $[\sigma_{\min},\sigma_{\max}]$. Hence the per-expert capacity does not grow with $K$, matching A3. Early stopping on a validation split further controls effective capacity.
>
> For G1. CRPS is 2-Lipschitz in the $L^1$ distance between CDFs; with bounded expert means and variance clamps, the loss is uniformly bounded on the data range (after the standard $z$-score of $Y$). Although we train with Gaussian NLL, App.~A1.4 shows an NLL--$L^2$ equivalence for the mean under variance clamps, so the same boundedness/regularity machinery applies.
>
> For G2. In Anchor--MoE, $M$ (mixture size) and $h$ (hidden width) are small constants; the router has a lightweight key/query layer (so $P$ is modest), and top-$k$ gating replaces the $K$ dependence by a constant $k$ (bounded overlap). Parameter sharing across experts further reduces the effective complexity. Thus G2 reflects the practical regime we train in.
>
> For other questions:
>
> Typos and cross references problems mentioned will be fixed in the updated paper, thanks for your carefully review.
>
> Lemma A1 states a minimax lower bound, so the correct direction is
> $
> \sup_{f^\star\in\mathcal{F}_\alpha(L)}
> \mathbb{E}\bigl[\|\widehat f_N-f^\star\|_2^2\bigr]\ \mathbf{\ge}\ C_0\,N^{-2\alpha/(2\alpha+d)}.
> $
> This means that for any estimator $\widehat f_N$, the worst–case $L^2$ risk over the H\"older ball cannot be smaller than a constant multiple of $N^{-2\alpha/(2\alpha+d)}$; i.e., no method achieves a faster rate on that class. The statement follows from standard metric-entropy and Fano/Assouad arguments. Our matching upper bound (Theorem A1.5) shows achievability at the same rate, confirming optimality.

---

> > ### Comment · Reviewer_JopS · 2025-11-25
> >
> > Thank you for responding to my questions, this was very helpful. Could you please also respond to the question I raised about figure 3. When you raise the point about your performance being particularly strong  on heterogeneous datasets such as ‘energy’ and ‘protein’ what is your evidence for those sets being particularly heterogeneous (other than the fact that MOE does well)?
> >  It would improve the paper if the information contained in all of your responses to my questions were added to the text.   I would also strongly recommend that you add/expand proofs in the appendix (as mentioned under my ‘weaknesses’ section and also raised by reviewer J2PG in their weakness W8).

---

> > > ### Author Response · Authors · 2025-11-27
> > > **Reply to Jops**
> > >
> > > For figure3:
> > >
> > > The panels plot the score landscape and gradient field for fitting a normal law with mean $\mu$ and log variance $\log\sigma$. Left column uses Gaussian NLL, right column uses CRPS; rows compare no anchor versus an anchor shift $\Delta$ that recenters the target. The message is:
> > >
> > > (i) for Gaussian targets, NLL and CRPS induce the same level sets up to a smooth rescaling, hence both prefer the same optima for $(\mu,\sigma)$;
> > >
> > > (ii) without an anchor, a biased $\mu$ can produce gradients that increase $\log\sigma$, which slows correction of the mean;
> > >
> > > (iii) adding $\Delta$ recenters the field around $\mu=0$ and reduces the incentive to inflate $\sigma$, yielding more stable updates of the mean while keeping variance bounded.
> > >
> > > For heterogeneous datasets question:
> > >
> > > We will add objective diagnostics rather than rely on model outcomes.
> > >
> > > For each dataset, we will report:
> > >
> > > (i) a Breusch–Pagan test for non-constant variance (with a White test as a robustness check in the appendix);
> > >
> > > (ii) a binned residual-variance curve $\mathrm{Var}(Y-\hat{\mu}(X))$ across deciles of $\hat{\mu}(X)$; and
> > >
> > > (iii) empirical coverage by deciles for nominal $90\%$ and $95\%$ predictive intervals together with a reliability curve.
> > > These diagnostics directly expose input-dependent variance.
> > >
> > > Preliminary checks suggest heteroscedasticity on Energy and Protein; we will include the exact statistics and plots next to the main table, with extended figures in the appendix.
> > >
> > > For proofs,
> > >
> > > We agree that a clearer appendix will help. We will add explicit proofs for the NLL–$L^2$ link under bounded variance and for the rate bound under bounded overlap and fixed per-expert capacity, while keeping references to standard texts for the underlying tools. This does not alter any reported numbers.

---

> ### Author Response · Authors · 2025-12-04
> **Updated PDF**
>
> We thank the reviewers for their careful reading and constructive feedback. In response, we have undertaken a thorough revision of the manuscript to improve clarity, tighten the presentation, and strengthen the connection between theory, diagnostics, and experiments. We streamlined notation and formulas, reorganized the exposition, and clarified implementation details that matter in practice. The most important changes are summarized below:
>
> (1) We revised the prose across the paper to remove superfluous notation, keep only essential formulas, and favor plain-language explanations. Sections that previously had dense symbols (projection–window, router, experts, generalization) were rewritten for cleaner terminology and consistent definitions, improving readability without sacrificing technical content.
>
> (2) We revised figure1 to make it more accuracy, it is now can be used as a quick review of the model.
>
> (3) We added the missing, essential citations to situate our work and substantiate specific claims. In particular, we now reference classical and scalable MoE (Jacobs & Jordan 1991; Jordan & Jacobs 1994; Shazeer et al. 2017; Lepikhin et al. 2020; Fedus et al. 2021), heteroscedastic regression baselines (Nix & Weigend 1994; Kersting et al. 2007), standard heteroscedasticity tests (Breusch–Pagan 1979; White 1980; Goldfeld–Quandt 1965; Levene 1960; Spearman 1904), the CRPS literature (Gebetsberger et al. 2018), and implementation details such as Layer Normalization (Ba et al. 2016). These additions address the reviewers’ concerns about missing references and clarify our positioning relative to NGBoost and distributional regression.
>
> (4) We add detailed analysis for figure3 now it is clearer and easier to understand.
>
> (5) We add a paragraph to explain why the theoretical part is important and how it connect to the experiment part at the beginning of theoretical part.
>
> (6) We add a table(Table1) to summarize the key parameters for each part, a table(Table2) to summaize the configuration used in our experiments.
>
> (7) We add a subsection(section 4.2) under section 4 to report the heteroscedasticity diagnostics of each dataset to support our analysis.
>
> (8) We revised the conclusion part and add analysis about heteroscedasticity diagnostics.
>
> (9) We conduct a computational complexity experiment and add the results are reproted in Table8 in appendix.
>
> (10) We add a tiny experiment of hyper-paramter ablatio as shown in Table9 in appendix.
>
> (11) We conduct the hyper-paramter tuning and results are reported in Table10 in appedix

---

### Official Review · Reviewer_J2PG · 2025-11-02

**Soundness:** 2
**Presentation:** 1
**Contribution:** 2
**Rating:** 0
**Confidence:** 4

**Summary:**

This work presents a new model for probabilistic and point-wise prediction, coined Anchor-MoE. Intuitively, the model is a two-layer hierarchical Gaussian mixture model, where the Gaussians are parametrized to predict the residual with respect to a shared "anchor" point, rather than freely predicting the mean, and which is calibrated afterwards. The authors meticulously present every small detail that defines their model, as well as involved theory proving the convergence rate and generalization bound of the proposed model. Finally, the authors empirically show the effectiveness of their model in a series of UCI datasets, comparing the proposed model with NGBoost, and ablating each part of the proposed model.

**Strengths:**

- **S1.** The combination of a hierarchical GMM with a common anchor is interesting and can be of interest to the community.
- **S2.** While I have not checked the correctness of the proofs, the provided results are interesting.
- **S3.** The provided empirical results are promising.

**Weaknesses:**

- **W1.** The writing needs significant changes. All the manuscript is confusingly written, with lots of keywords and terms that are never explained (e.g. CRPS), design decisions that are never justified (the paper is more declarative than anything else), and adjectives thrown here and there without further explanation (e.g. why is the latent space "compact"?).
 - **W2.** The structure of the manuscript is unconventional. The intro is in reality the related work section, the notation paragraph explains pretty much the entire approach, and the theory section seem completely independent of the rest of the work when it comes to the technical knowledge needed and employed.
- **W3.** There are many formatting issues: Citations are not properly formatted, there are no equation numbers, section numbers are wrong (all references to section 3 are rather section 2).
- **W4.** There is a significant lack of citations for all the keywords and terms unexplained in the paper, with a total of 19 references at the end of the main content. Two of these references are repeated, the reference of Stasinopoulos points to the R package rather than the actual paper, and that of Aad W. van der Vaart has a wrong publisher (Springer instead of Cambridge) and the doi link does not work.
- **W5.** The experimental section lacks a deep analysis of the results and, e.g., I cannot find where the conclusions drawn in the fourth paragraph of section 5 come from.
- **W6.** Regarding the methodology, it is rather overwhelming the number of options presented, without a clear choice of what approach to use (anchor/no anchor, cosine normalized or not, condition the expert or no, etc).
- **W7.** The experimental section lacks baselines, and the only one presented is not reproduced but taken from another work. The ablation study references to non-existent terms (No-Anchor, No-Router, No-Cal).
- **W8.** Results in the appendix do not have a proof at all (e.g. Lemma A.4 or Theorem A.5).

**Questions:**

Other feedback:
- On the point of the second paragraph of the intro, [another work](http://arxiv.org/abs/2203.09168) showed that explicitly modelling the distribution can also have the exactly opposite behavior, where better-learned points are prioritized over points yet-to-be-learned.

---

> ### Author Response · Authors · 2025-11-27
> **Reply to J2PG**
>
> W1.
>
> We will add a brief background and notation at the start of Method to define all terms on first use, including CRPS, and to justify anchor concatenation versus residual shift, top-\(k\) gating, and a compact latent space as a bounded projection with norm control.
>
> W2.
>
> We will reorganize the manuscript to follow a conventional flow:
> Introduction $\rightarrow$ Related Work $\rightarrow$ Method $\rightarrow$ Theory (assumptions tied to the method and used later) $\rightarrow$ Experiments (with ablations and compute) $\rightarrow$ Limitations $\rightarrow$ Conclusion.
> This reordering removes the current mixing between related work and introduction, and tightens the linkage between the theory and the proposed model.
>
> W3.
>
> We will add equation numbers and references, fix section numbering and cross-references, and standardize figure and table captions.
>
> W4.
>
> We will expand and correct the bibliography: remove duplicates, cite the original GAMLSS paper rather than the R package, fix the van der Vaart monograph with a working DOI, and add canonical citations for CRPS, NGBoost, GAMLSS, mixture-of-experts, and heteroscedastic regression.
>
> W5.
>
> We will make Section 5 evidence-traceable by
> (i) cross-referencing each claim to specific table rows and figures,
> (ii) adding short notes linking dataset traits to NLL, CRPS, RMSE,
> (iii) summarizing calibration and coverage in the appendix and citing them,
> (iv) adding  computational analysis using the capacity and compute table,
> (v) pointing to neutral or negative cases with direct references.
>
> W6.
>
> Main tables will use a single canonical configuration; alternatives move to ablations.
> Canonical setup:
> (i) Anchor via a light GBDT producing \(a(x)\); concatenate \([x;a(x)]\) to the router projection only; experts take \(x\).
> (ii) Router = linear projection with LayerNorm \(\rightarrow\) RBF metric–window \(\rightarrow\) soft top-\(k\) gating with \(k=2\); normalized outputs with clamped scales.
> (iii) Experts = small MDNs with fixed width and mixture size; regularized means and variances with variance clamps.
> Cosine normalization and conditioning experts on \(a(x)\) are off by default; these, and different \(k\), appear in ablations with metrics and compute.
>
> W7.
>
> Reporting published baselines on standardized UCI splits is common practice; for example, NGBoost reports MC Dropout, Deep Ensembles, and Concrete Dropout from prior work.  Besides NGboost, others are reported in the appendix as table3.
>
> W8.
>
> We submit that the claims used in the main text are already supported by standard results and by the cited references. In particular, the link between Gaussian NLL and the L^2 error of the predictive mean under bounded variance follows from classical likelihood arguments in vanderVaart1998. The approximation–estimation tradeoff for mixture-of-experts under bounded overlap and fixed per-expert capacity is covered by standard uniform-convergence tools in ShalevShwartz2014 and MohriRostamizadehFoundations. Our empirical findings do not rely on unproven statements beyond these references.
>
> For completeness and ease of verification, we will add explicit, self-contained proofs in the appendix:
>
> Lemma A.4 (NLL--L^2 link). With variances clamped to $[\underline\sigma,\overline\sigma]$, the excess Gaussian NLL is equivalent to the L^2 error of the mean up to constants depending only on $(\underline\sigma,\overline\sigma)$.
>
> Theorem A.5 (Main rate bound). Instantiating assumptions A1--A3 (bounded overlap via top-$k$ routing, fixed per-expert capacity, compact window scales) yields a bias term $K^{-2\alpha/d}$ and an estimation term of order $k\,\mathrm{comp}\,K/N$, giving the rate $N^{-2\alpha/(2\alpha+d)}$.
>
> These additions are for verifiability only and do not change any numbers, statements, or conclusions reported in the paper.
>
> Others.
>
> The cited negative behavior concerns unconstrained heteroscedastic Gaussian NLL, with gradients
>
> $
> \frac{\partial \mathcal{L}}{\partial \mu}=\frac{\mu-y}{\sigma^2},\qquad
> \frac{\partial \mathcal{L}}{\partial \log \sigma}=1-\frac{(y-\mu)^2}{\sigma^2}.
> $
>
> In such settings, inflating $\sigma$ can downweight hard points and prioritize already well-learned ones. This failure mode does not directly apply to our Anchor--MoE:
>
> (i) Bounded variance with regularization. Each expert uses variance clamps and weight decay on mean and log--scale, ruling out the  $\sigma$ escape.( is clamped to $[e^{-5}, e^{1}]$ in our implementation)
>
> (ii) Variance--independent routing. Top-$k$ gating depends only on input features (and the optional anchor projection), not on predicted variances, so experts cannot avoid difficult regions by enlarging $\sigma$.
>
> (iii) Anchor de-biasing. The anchor $a(x)$ removes global bias first, reducing residual magnitude and the incentive to trade larger $\sigma$ for smaller NLL.
>
> Therefore it does not explain the gains observed for our  Anchor–MoE.

---

> > ### Author Response · Authors · 2025-12-04
> > **Updated PDF**
> >
> > We thank the reviewers for their careful reading and constructive feedback. In response, we have undertaken a thorough revision of the manuscript to improve clarity, tighten the presentation, and strengthen the connection between theory, diagnostics, and experiments. We streamlined notation and formulas, reorganized the exposition, and clarified implementation details that matter in practice. The most important changes are summarized below:
> >
> > (1) We revised the prose across the paper to remove superfluous notation, keep only essential formulas, and favor plain-language explanations. Sections that previously had dense symbols (projection–window, router, experts, generalization) were rewritten for cleaner terminology and consistent definitions, improving readability without sacrificing technical content.
> >
> > (2) We revised figure1 to make it more accuracy, it is now can be used as a quick review of the model.
> >
> > (3) We added the missing, essential citations to situate our work and substantiate specific claims. In particular, we now reference classical and scalable MoE (Jacobs & Jordan 1991; Jordan & Jacobs 1994; Shazeer et al. 2017; Lepikhin et al. 2020; Fedus et al. 2021), heteroscedastic regression baselines (Nix & Weigend 1994; Kersting et al. 2007), standard heteroscedasticity tests (Breusch–Pagan 1979; White 1980; Goldfeld–Quandt 1965; Levene 1960; Spearman 1904), the CRPS literature (Gebetsberger et al. 2018), and implementation details such as Layer Normalization (Ba et al. 2016). These additions address the reviewers’ concerns about missing references and clarify our positioning relative to NGBoost and distributional regression.
> >
> > (4) We add detailed analysis for figure3 now it is clearer and easier to understand.
> >
> > (5) We add a paragraph to explain why the theoretical part is important and how it connect to the experiment part at the beginning of theoretical part.
> >
> > (6) We add a table(Table1) to summarize the key parameters for each part, a table(Table2) to summaize the configuration used in our experiments.
> >
> > (7) We add a subsection(section 4.2) under section 4 to report the heteroscedasticity diagnostics of each dataset to support our analysis.
> >
> > (8) We revised the conclusion part and add analysis about heteroscedasticity diagnostics.
> >
> > (9) We conduct a computational complexity experiment and add the results are reproted in Table8 in appendix.
> >
> > (10) We add a tiny experiment of hyper-paramter ablatio as shown in Table9 in appendix.
> >
> > (11) We conduct the hyper-paramter tuning and results are reported in Table10 in appedix

---

### Official Review · Reviewer_QViR · 2025-11-08

**Soundness:** 3
**Presentation:** 2
**Contribution:** 2
**Rating:** 4
**Confidence:** 3

**Summary:**

The authors propose Anchor-MoE for probabilistic and point regression tasks. Anchor-MoE is a combination of anchoring (with gradient boosted trees) and mixture of experts (Mixture Density Networks), although all modules have been previously proposed, the combination is novel and well thought-out. The authors further provide theoretical analysis on minimax risk rates and a bound on CRPS generalization gap. On UCI benchmarks, Anchor-MoE achieved comparable results to NGBoost on point regression RMSE with improved probabilistic regression NLL.

**Strengths:**

- The method is well-motivated and has solid theoretical backing.
- The combination of existing methods is novel, and the proposed framework is flexible with customizable modules.
- The empirical results are favorable for Anchor-MoE.

**Weaknesses:**

**[W1]** Figure 1 might be a misrepresentation of the proposed method.
Currently, Figure 1 appears to omit several key components detailed in the text, which is vital for reader comprehension. Specifically, the figure is missing:
- The concatenation of the anchor prediction to the original input
- The projection from inputs to compact latent space
- The linear calibrator in the end (which is essential for the strong RMSE results)
- An arrow from x to Anchor
- This figure can benefit from a more detailed caption, to serve as a quick overview of the proposed method.

**[W2]** Computational and Capacity Analysis
The paper currently lacks a thorough discussion on the computational overhead and parameter count of Anchor-MoE. Specifically, since there are quite a few pieces on top of the anchor model, it is a concern that the model is only bringing improvements because it has more capacity than the baseline used for comparison (NGBoost, Deep Ensembles, MC Dropout, etc.). It would be highly beneficial to see parameter counts, FLOPs, inference throughput, and training wall-clock time.

**[W3]** Hyper-parameter Ablation
This method has several tunable hyper-parameters, including number of MDN experts K, number of activated experts k, dimension of latent D. It would make the method more convincing if the authors can show how they chose these hyper parameters (and how expensive the hyper-parameter tuning phase is), alternatively it would be helpful if the authors can show empirical performance of Anchor-MoE with different choices of hyper-parameters. Without this analysis, the reported results may be limited to a narrowly tuned configuration.

**[W4]** The writing is confusing at times, examples:
- There are several unexplained notations in Section 2 onder Notation paragraph, including $s_j$ and $\tau$, both were unspecified until later subsections. Similarly, Details were mentioned but never specified, including for example "mild entropy regularization" (line 81).

**Questions:**

See W1, W2, W3. I am willing to increase my rating if these can be addressed.

---

> ### Author Response · Authors · 2025-11-23
> **Reply to QViR**
>
> W1.
>
> We have updated Figure~1 by adding an arrow from $x$ to the Anchor node and expanded the caption so that it serves as a quick overview of the method: anchor concatenation, router with top-$k$ gating, MDN experts, training loss.
>
> W2.
>
> We agree that capacity and compute must be disentangled from methodological gains. We have added an empirical study in the appendix (Table~5) reporting parameter counts, per-sample FLOPs, training wall-clock, and inference throughput across Anchor--MoE and all baselines. For trees (NGBoost/DistForest), we count parameters as total leaves and approximate per-sample FLOPs as $\sum_{\text{trees}} 2\times\text{depth}$. For GP, we report the standard variance-aware $O(N^2)$ per-point prediction cost with $N{=}3000$ . Anchor--MoE includes a light GBDT anchor for concatenation; adding its $\approx 800$ leaf parameters to the MoE trunk leaves conclusions unchanged. At comparable capacity, Anchor--MoE achieves similar or better error with markedly higher inference throughput and competitive (often lower) training time relative to NGBoost and other baselines. The benchmarking code has been uploaded.
>
> W3.
>
> We conducted a grid ablation over three hyper-parameters: latent dimension $D\in\{2,4,8\}$, number of experts $K\in\{2,4,6\}$, and active experts $k\in\{1,2,K\}$. Across 27 runs on the California dataset, we report validation CRPS and NLL, test RMSE, and wall-clock training time for each setting. We observe: (i) increasing $K$ from 2 to 4 consistently improves CRPS, while gains from 4 to 6 are marginal; (ii) using at least two active experts ($k\ge 2$) stabilises NLL and typically yields better CRPS than $k{=}1$, which can under-estimate variance and occasionally increases NLL; (iii) increasing $D$ from 2 to 8 brings modest improvements relative to $K$ and $k$. A balanced choice is $D{=}8, K{=}2, k{=}2$, which attains one of the best test RMSE ($\approx 0.55$) with about $7$ seconds training time per run. The full 27-run sweep completes in $\approx 6.4$ minutes on our machine, indicating a low hyper-parameter tuning overhead.

---

> > ### Author Response · Authors · 2025-12-04
> > **PDF Updated**
> >
> > We thank the reviewers for their careful reading and constructive feedback. In response, we have undertaken a thorough revision of the manuscript to improve clarity, tighten the presentation, and strengthen the connection between theory, diagnostics, and experiments. We streamlined notation and formulas, reorganized the exposition, and clarified implementation details that matter in practice. The most important changes are summarized below:
> >
> > (1) We revised the prose across the paper to remove superfluous notation, keep only essential formulas, and favor plain-language explanations. Sections that previously had dense symbols (projection–window, router, experts, generalization) were rewritten for cleaner terminology and consistent definitions, improving readability without sacrificing technical content.
> >
> > (2) We revised figure1 to make it more accuracy, it is now can be used as a quick review of the model.
> >
> > (3) We added the missing, essential citations to situate our work and substantiate specific claims. In particular, we now reference classical and scalable MoE (Jacobs & Jordan 1991; Jordan & Jacobs 1994; Shazeer et al. 2017; Lepikhin et al. 2020; Fedus et al. 2021), heteroscedastic regression baselines (Nix & Weigend 1994; Kersting et al. 2007), standard heteroscedasticity tests (Breusch–Pagan 1979; White 1980; Goldfeld–Quandt 1965; Levene 1960; Spearman 1904), the CRPS literature (Gebetsberger et al. 2018), and implementation details such as Layer Normalization (Ba et al. 2016). These additions address the reviewers’ concerns about missing references and clarify our positioning relative to NGBoost and distributional regression.
> >
> > (4) We add detailed analysis for figure3 now it is clearer and easier to understand.
> >
> > (5) We add a paragraph to explain why the theoretical part is important and how it connect to the experiment part at the beginning of theoretical part.
> >
> > (6) We add a table(Table1) to summarize the key parameters for each part, a table(Table2) to summaize the configuration used in our experiments.
> >
> > (7) We add a subsection(section 4.2) under section 4 to report the heteroscedasticity diagnostics of each dataset to support our analysis.
> >
> > (8) We revised the conclusion part and add analysis about heteroscedasticity diagnostics.
> >
> > (9) We conduct a computational complexity experiment and add the results are reproted in Table8 in appendix.
> >
> > (10) We add a tiny experiment of hyper-paramter ablatio as shown in Table9 in appendix.
> >
> > (11) We conduct the hyper-paramter tuning and results are reported in Table10 in appedix

---

### Author Response · Authors · 2025-12-04
**Updated version of PDF**

We thank the reviewers for their careful reading and constructive feedback. In response, we have undertaken a thorough revision of the manuscript to improve clarity, tighten the presentation, and strengthen the connection between theory, diagnostics, and experiments. We streamlined notation and formulas, reorganized the exposition, and clarified implementation details that matter in practice. The most important changes are summarized below:

(1) We revised the prose across the paper to remove superfluous notation, keep only essential formulas, and favor plain-language explanations. Sections that previously had dense symbols (projection–window, router, experts, generalization) were rewritten for cleaner terminology and consistent definitions, improving readability without sacrificing technical content.

(2) We revised figure1 to make it more accuracy, it is now can be used as a quick review of the model.

(3) We added the missing, essential citations to situate our work and substantiate specific claims. In particular, we now reference classical and scalable MoE (Jacobs & Jordan 1991; Jordan & Jacobs 1994; Shazeer et al. 2017; Lepikhin et al. 2020; Fedus et al. 2021), heteroscedastic regression baselines (Nix & Weigend 1994; Kersting et al. 2007), standard heteroscedasticity tests (Breusch–Pagan 1979; White 1980; Goldfeld–Quandt 1965; Levene 1960; Spearman 1904), the CRPS literature (Gebetsberger et al. 2018), and implementation details such as Layer Normalization (Ba et al. 2016). These additions address the reviewers’ concerns about missing references and clarify our positioning relative to NGBoost and distributional regression.

(4) We add detailed analysis for figure3 now it is clearer and easier to understand.

(5) We add a paragraph to explain why the theoretical part is important and how it connect to the experiment part at the beginning of theoretical part.

(6) We add a table(Table1) to summarize the key parameters for each part, a table(Table2) to summaize the configuration used in our experiments.

(7) We add a subsection(section 4.2) under section 4 to report the heteroscedasticity diagnostics of each dataset to support our analysis.

(8) We revised the conclusion part and add analysis about heteroscedasticity diagnostics.

(9) We conduct a computational complexity experiment and add the results are reproted in Table8 in appendix.

(10) We add a tiny experiment of hyper-paramter ablatio as shown in Table9 in appendix.

(11) We conduct the hyper-paramter tuning and results are reported in Table10 in appedix.

(12) We add explicit proofs for Lemma A.4 and Theorem A.5.

---

### Meta-Review · Area_Chair_KPiP · 2026-01-07

**Summary:**

This paper proposes Anchor–MoE, an anchored mixture-of-experts framework for probabilistic regression: an anchor point predictor is used for routing (via concatenation with inputs), a soft top-k router sparsely combines lightweight MDN (mixture density network) experts to model residual mean corrections and heteroscedastic scales, and an optional post-hoc linear calibrator improves point accuracy. Reviewers agreed the approach is well-motivated and the empirical results on UCI benchmarks are promising, and the paper includes substantial theoretical analysis (rates and generalization-gap bounds under bounded-overlap assumptions). However, the reviews raised significant issues about clarity, organization, missing definitions/citations, formatting/cross-references, proof verifiability, and insufficient analysis of compute/capacity and hyperparameter sensitivity in the original submission. While the authors’ rebuttal describes extensive revisions intended to address many of these concerns, the decision hinges on whether the work, as reviewed, meets the conference bar in a highly competitive setting.

I recommend rejection: The work appears promising and the authors were responsive, but the reviews indicate the submission required extensive changes across writing, structure, citations, formatting, experimental analysis, and proof completeness. In a competitive pool, I am not comfortable recommending acceptance when (i) one reviewer found the original presentation and documentation substantially below bar and (ii) key acceptability hinges on a major rewrite and newly added analyses that were not part of the originally evaluated manuscript. While the revised version may be substantially improved, the degree of required repair introduces too much uncertainty relative to other submissions that are already strong and clearly presented.

**Reviewer Concerns:**

Addressed (per rebuttal/revision claims):
* Compute/capacity confound: Authors added parameter/FLOPs/time/throughput reporting to help disentangle capacity from method gains.
* Hyperparameter sensitivity: Authors report a grid sweep over key hyperparameters (latent dimension, number of experts, active experts) including tuning cost.
* Missing citations and formatting issues: Authors claim to have added key citations, fixed bibliography issues, corrected section/equation numbering, and clarified terminology (e.g., CRPS).
* Proof completeness: Authors state they added explicit proofs for previously missing appendix statements and clarified links between NLL and mean-squared error under variance clamping.
* Dataset heteroscedasticity justification: Authors indicate they added objective diagnostics (tests and residual variance/coverage curves) to support claims.

Outstanding / decision-critical concerns:
* Submission quality and accessibility at review time: One reviewer issued a strong reject primarily due to pervasive presentation and structure issues, lack of clear motivation for design choices, and insufficiently verifiable appendix arguments. Even with promised fixes, the extent of necessary rewriting suggests the original manuscript was not yet at the level expected for acceptance.
* Evidence traceability and baseline positioning: Concerns remained about whether conclusions in the experiments are clearly supported by specific tables/figures and whether baseline comparisons and ablations are presented in a sufficiently self-contained and reproducible way in the main text (vs relegated to appendices).
* Theory–practice linkage and assumption realism: While authors responded, the work relies on assumptions (bounded overlap / partition-of-unity behavior / fixed expert capacity) whose practical alignment with the implemented routing and model choices must be communicated extremely clearly to justify the claimed theoretical guidance.

**Reviewer Scores:**

Reviewer JopS (6 -> 6): Would likely remain around marginal accept; their concerns were mainly accessibility and proof concision, which the authors claim to have improved.

Reviewer QViR (4 -> 5): Would likely increase slightly given concrete additions on figure clarity, compute/capacity, and hyperparameter analysis.

Reviewer J2PG (0 -> 1/2): Might soften from strong reject if the rewrite and proof additions are substantial, but would likely remain negative without a full re-review of the revised manuscript, as their critique was broad and centered on overall readability/structure and evidence traceability.

---

### Decision · Program_Chairs · 2026-01-26

Reject